# Forecasting protein evolution by integrating birth-death population models with structurally constrained substitution models

**David Ferreiro[1,2], Luis Daniel González-Vázquez[1,2], Ana Prado-Comesaña[1], Miguel Arenas[1,2]***

[1]CINBIO, Universidade de Vigo, Vigo, Spain; [2]Department of Biochemistry, Genetics and Immunology, Universidade de Vigo, Vigo, Spain

**\*For correspondence:**
marenas@uvigo.es

**Competing interest:** The authors declare that no competing interests exist.

## eLife Assessment

This manuscript introduces a **useful** protein-stability-based fitness model for simulating protein evolution and unifying non-neutral models of molecular evolution with phylogenetic models. The model is applied to five viral proteins that are of structural and functional importance. While the general modelling approach is **solid**, and effectively preserves folding stability, the evidence for the model's predictive power remains limited, since it shows little improvement over neutral models in predicting protein evolution. The work should be of interest to researchers developing theoretical models of molecular evolution.

**Abstract** Evolutionary studies in population genetics and ecology were mainly focused on predicting and understanding past evolutionary events. Recently, however, a growing trend explores the prediction of evolutionary trajectories toward the future promoted by its wide variety of applications. In this context, we introduce a forecasting protein evolution method that integrates birth-death population models with substitution models that consider selection on protein folding stability. In contrast to traditional population genetics methods that usually make the unrealistic assumption of simulating molecular evolution separately from the evolutionary history, the present method combines both processes to simultaneously model forward-in-time birth-death evolutionary trajectories and protein evolution under structurally constrained substitution models that outperformed traditional empirical substitution models. We implemented the method into a freely available computer framework. We evaluated the accuracy of the predictions with several monitored viral proteins of broad interest. Overall, the method showed acceptable errors in predicting the folding stability of the forecasted protein variants, but, expectedly, the errors were larger in the prediction of the corresponding sequences. We conclude that forecasting protein evolution is feasible in certain evolutionary scenarios and provide suggestions to enhance its accuracy by improving the underlying models of evolution.

## Introduction

Molecular evolution is traditionally investigated through inferences about past evolutionary events, such as phylogenetic tree and ancestral sequence reconstructions, and predictions about the future were considered as inaccessible for a long time because they can be affected by complex processes such as environmental change. Nevertheless, a variety of biological systems display a Darwinian

evolutionary process where selection operates toward a limited set of adapted variants. These variants, and in extension the evolutionary trajectories to reach them, would be positively selected and could present a certain degree of predictability (*Lässig et al., 2017*; *Wortel et al., 2023*). The progress made in developing more accurate models of evolution (*Arenas, 2015b*) and the benefits from predicting the outcome of evolution (i.e. to understand the course of evolution or to prepare for the future *Wortel et al., 2023*) motivated a variety of investigations on forecasting evolution in diverse fields including medicine, agriculture, biotechnology, and conservation biology, among others (e.g. *Barton et al., 2016*; *Bull and Molineux, 2008*; *de Visser et al., 2018*; *Diaz-Uriarte and Vasallo, 2019*; *Fischer et al., 2015*; *Gerrish and Sniegowski, 2012*; *Lässig and Łuksza, 2014*; *Lind et al., 2019*; *Luksza and Lässig, 2014*; *Morris et al., 2018*; *Munck et al., 2014*; *Neher et al., 2014*; *Wortel et al., 2023*). Unfortunately, forecasting evolution is not always achievable. Under neutral evolution, all the molecular variants are equally likely to be present in the population, showing lack of repeatability and disallowing accurate prediction of future variants. Thus, forecasting evolution requires a system with measurable selection pressures, and where certain positively selected variants could produce more descendants than other variants and expand in the population (*Desai and Fisher, 2007*; *Goyal et al., 2012*; *Neher and Hallatschek, 2013*; *Neher et al., 2014*). Actually, a rougher fitness landscape resulting from selection can lead to greater accuracy in evolutionary predictions (*Papkou et al., 2023*; *Rubin et al., 2023*; *Van Cleve and Weissman, 2015*). Overall, an evolutionary process could be predictable to some extent (prediction errors are inevitable with any method and in any evolutionary scenario) depending on the strength of selection driving evolution and the heterogeneity in fitness among different variants.

Here, we focus on forecasting protein evolution because it involves molecular evolutionary processes driven by selection pressures and where the fitness of each variant can be parameterized and predicted (*Carneiro and Hartl, 2010*; *Gilson et al., 2017*). Traditionally, evolutionary histories and ancestral sequences of proteins are inferred using probabilistic methods based on advanced substitution models of protein evolution (e.g. *Arenas, 2015b*; *Arenas and Bastolla, 2020*; *Ferreiro et al., 2022*; *Malcolm et al., 1990*; *Moreira et al., 2023*; *Stackhouse et al., 1990*; *Thornton et al., 2003*; *Ugalde et al., 2004*). The accuracy of these inferences is affected by the accuracy of the applied substitution model, where substitution models that better fit with the study data usually produce more accurate phylogenetic trees and ancestral sequences (*Arenas and Bastolla, 2020*; *Del Amparo and Arenas, 2022a*; *Lemmon and Moriarty, 2004*). These findings suggest that accurate substitution models of evolution are also convenient for forecasting protein evolution. In this regard, a variety of studies showed that structurally constrained substitution (SCS) models of protein evolution provide more accurate evolutionary inferences than the traditional empirical substitution models of protein evolution, in terms of phylogenetic likelihood, distribution of amino acid frequencies among protein sites, rates of molecular evolution and folding stability of reconstructed proteins, among other aspects (e.g. *Arenas and Bastolla, 2020*; *Arenas et al., 2016a*; *Bastolla et al., 2006*; *Bordner and Mittelmann, 2014*; *Del Amparo et al., 2023*; *Echave and Wilke, 2017*; *Ferreiro et al., 2024a*; *Ferreiro et al., 2024b*; *Fornasari et al., 2002*; *Parisi and Echave, 2001*; *Pascual-García et al., 2019*; *Rodrigue et al., 2005*), although SCS models usually demand more computational resources than substitution models that only include information from the protein sequence. Notice that the protein structure provides information about the location and molecular interactions of amino acids at different protein sites, which could be far from each other in the sequence but close in the three-dimensional structure and interact affecting their evolution (*Morcos et al., 2011*; *Ruiz-González and Fares, 2013*). Indeed, selection from the protein folding stability is relevant in the evolution of multiple proteins, including those in microbial and viral systems (e.g. *Ferreiro et al., 2022*; *Gong et al., 2013*; *Jacquier et al., 2013*; *Rodrigues et al., 2016*; *Wylie and Shakhnovich, 2011*; *Zeldovich et al., 2007*). Therefore, we believe that it should be taken into account for forecasting protein evolution in such systems.

Predictions about future evolutionary events can be performed with simulation-based methods (e.g. *Eccleston et al., 2023*; *Neher et al., 2014*; *Yoshida et al., 2023*). In order to simulate molecular evolution, traditional population genetics methods apply two separate steps (*Arenas, 2012*; *Hoban et al., 2012*). First, the simulation of the evolutionary history (i.e. a phylogenetic tree) using approaches such as the coalescent and birth-death population processes (*Gernhard, 2008*; *Hudson, 1990*; *Kingman, 1982*; *Stadler, 2010*). Afterward, the forward-in-time simulation of molecular evolution is performed, from the root node to the tip nodes, upon the previously simulated evolutionary

history (*Yang, 2006*). This methodology was implemented into a variety of population genetics frameworks that simulate molecular evolution (*Arenas, 2012*; *Hoban et al., 2012*). However, for technical and computational simplicity, it assumes that the simulation of the evolutionary history is independent from the simulation of molecular evolution, which can produce biological incoherences (i.e. the evolutionary history is usually simulated under neutral evolution while molecular evolution is usually simulated with substitution models that consider selection). To enhance the realism of this modeling, here we merged both processes into a single one where evolutionary history influences molecular evolution and *vice versa*. In particular, we adopted a birth-death population genetics method to simulate the forward-in-time evolutionary history already used for forecasting evolution (*Lässig and Łuksza, 2014*; *Neher et al., 2014*), taking into account the fitness of the molecular variant (through evolutionary constraints from the protein folding stability *Bastolla and Demetrius, 2005*; *Gong et al., 2013*; *Liberles et al., 2012*; *Zeldovich et al., 2007*) at the corresponding node, to determine its subsequent birth or death event, and we integrated this process with SCS models to model protein evolution along the derived phylogenetic branches. The method is detailed below, and we implemented it into a new version of our computer framework *ProteinEvolver* (*Arenas et al., 2013*), which is freely available from https://github.com/MiguelArenas/proteinevolver (*Arenas, 2025*) *ProteinEvolver2* includes detailed documentation and a variety of ready-to-use examples. Next, considering the potential applications of forecasting evolution to design vaccines and therapies against pathogens, we evaluated and applied the method to forecasting protein evolution in several real protein data of viruses monitored over time.

## Methods
### A method for forecasting protein evolution by combining birth-death population genetics with structurally constrained substitution models of protein evolution

Following previous methods for forecasting evolution based on simulations (*Lässig and Łuksza, 2014*; *Neher et al., 2014*), we developed a method to simulate the forward-in-time evolutionary history of a protein sample with a birth-death process that considers the fitness of the protein variant (based on folding stability) at every temporal node. The method derives the birth and death rates for a protein variant based on the fitness of that variant, where a high fitness results in a high birth rate and a low death rate, which can lead to a large number of descendants, and the opposite leading to a few or none (extinction) descendants. Thus, the fitness of the molecular variant at every node drives its corresponding forward in time birth-death evolutionary history. The details of this simulation process are outlined below.

First, similarly to common simulators of molecular evolution (*Arenas, 2012*; *Hoban et al., 2012*; *Yang, 2006*), a given protein sequence and structure (hereafter, protein variant) is assigned to the root node. The fitness (*f*) of the protein variant (*A*) is calculated from its folding stability (free energy, *ΔG*) following the Boltzmann distribution (*Goldstein, 2013*) (*Equation 1*, which takes values from 0 to 1),

$$f(A) = \frac{1}{1 + e^{\Delta G/kT}} \tag{1}$$

Protein folding stability constrains protein evolution and is commonly used to obtain protein fitness (*Bastolla et al., 2007*; *Goldstein, 2013*; *Liberles et al., 2012*; *Lobkovsky et al., 2010*; *Mendez et al., 2010*; *Sella and Hirsh, 2005*; *Zeldovich et al., 2007*). The user can alternatively choose whether the fitness of the modeled protein variant is determined solely by its folding stability or by its similarity to the stability of a real protein variant (i.e., a protein structure from the Protein Data Bank, PDB). We believe the latter can be more realistic, as in nature, high folding stability does not necessarily indicate high fitness, but a stability that closely resembles that of a real protein may suggest high fitness since the real stability is the result of a selection process (which also incorporates negative design). If the fitness is derived from only the folding stability of the protein variant, the birth rate (*b*) is considered equal to the fitness. Alternatively, if the fitness is determined based on the similarity in folding stability between the modeled variant and a real variant, the birth rate is assumed to be 1 minus the root mean square deviation (RMSD, which offers advantages such as minimizing the influence of small deviations while amplifying larger differences, thereby enhancing the detection of remarkable

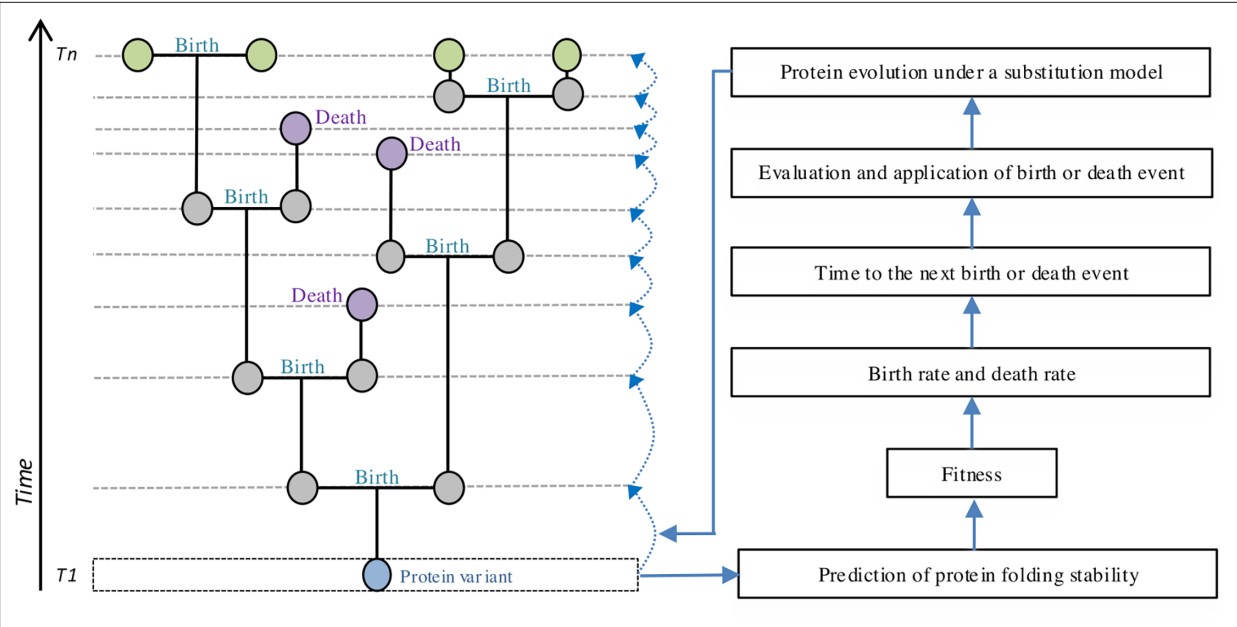

**Figure 1.** Illustrative example of forward in time simulation of protein evolution integrating a birth-death population evolutionary process with fitness from the protein folding stability and the modeling of protein evolution with a structurally constrained substitution model. Given a protein variant assigned to a node at time *t* (blue node), its fitness is calculated considering its protein folding stability. Then, the fitness is used to determine the birth and death rates for that variant, which provide the time to the next birth or death event (horizontal dashed line) that corresponds to the forward-in-time branch length. Next, the variant is evolved forward in time toward each descendant, upon the previously determined branch length, under an SCS model of protein evolution. The process is repeated, forward in time, starting at each new variant. If a death event occurs, the variant of the extinct node (pink node) is obtained, but it does have descendants. The process finishes when a particular sample size or simulation time is reached (i.e. *t+n*).

molecular changes) in folding stability. Notice that the smaller this difference, the higher the birth rate. In both cases, the death rate (*d*) is considered as *1-b* to allow a constant global (birth-death) rate. In this model, the fitness influences reproductive success, where protein variants with higher fitness have higher birth rates leading to more birth events, while those with lower fitness have higher death rates leading to more extinction events. This parameterization is meaningful in the context of protein evolution because the fitness of a protein variant can affect its survival (birth or extinction) without necessarily altering its rate of evolution. Although a higher growth rate can sometimes correlate with higher fitness, a variant with high fitness does not necessarily accumulate substitutions more rapidly.

Additionally, we incorporated another birth-death model that follows the proposal by *Neher et al., 2014*, in which the death rate is fixed at 1 and the birth rate is modeled as *1+fitness*. In this model, fitness not only affects reproductive success but also influences the global birth-death rate, which can vary among lineages.

The birth-death process is simulated forward in time deciding whether every next event is a birth or a death event (*Harmon, 2019*; *Stadler, 2010*; *Stadler, 2011*) according to the fitness of the corresponding molecular variant, and it ends when a user-specified criterion, such as a particular sample size (considering or ignoring extinction nodes) or a certain evolutionary time ($t_e$), is reached. Starting from an 'active' node (i.e. the root node) at current time $t_c$, the time to the next event (birth to produce two descendants or death to produce the extinction of the node) can be calculated (details below). In contrast to standard birth-death processes where birth and death rates are constant over time and among lineages, the present method considers heterogeneity where each protein variant at a node has specific birth and death rates according to its corresponding fitness. The method is described below and summarized in *Figure 1*.

(1) The process starts at the root node, assigning a user-specified protein sequence and corresponding protein structure (i.e. obtained from the PDB) to that node. In general, for every protein variant assigned to an active node, the corresponding birth and death rates are calculated following the indications presented above.

(2) Calculation of the time to the next birth or death event. Following common birth-death methods, the time to the next event $t_n$ is calculated through an exponential distribution with rate based on the number of active nodes (s) and the sum of the birth and death rates (**Harmon, 2019**; **Equation 2**),

$$t_n = e^{(s(b+d))} \tag{2}$$

One of the birth-death models that we implemented considers that *b+d = 1* at each node, allowing variation of the reproductive success among nodes while keeping $t_n$ consistent among them, according to **Harmon, 2019**. In contrast, in the other birth-death model we implemented, *b+d* can vary among nodes (**Neher et al., 2014**), thereby allowing variation of both reproductive success and $t_n$ among nodes.

(3) Evaluate whether the simulation concludes before the next event occurs ($t_c + t_n$).

(4) If it does conclude (i.e. $t_c + t_n$ higher than $t_e$), the simulation of the evolutionary history finishes.

(5) If it does not conclude, a random active node is selected, and its protein variant is analyzed to determine its type of next event (birth or death). The probability of a birth event is $P_b = b/(b+d)$ and the probability of a death event is $P_d = d/(b+d)$. A random sample from those probabilities is taken to determine the type of evolutionary event.

(6) If a birth event is selected. Two descendant active nodes from the study node are incorporated with branch lengths $t_n$, and the study node is then considered inactive. Next, molecular evolution is simulated from the original node to each descendant node based on the specified SCS model of protein evolution (**Arenas et al., 2013**). The integration of SCS models to evolve protein variants along given branch lengths followed standards approaches of molecular evolution in population genetics, in which the branch length and the substitution model inform the number and type of substitution events, respectively (**Arenas, 2012**; **Carvajal-Rodríguez, 2010**; **Hoban et al., 2012**; **Yang, 2006**). Thus, the process results in a protein variant for every descendant node. Finally, the folding stability and subsequent fitness of these descendant protein variants are calculated.

(7) If a death event is selected. Then, the study node is considered inactive.

(8) Return to step 1 while the user-specified criterion for ending the birth-death process is not satisfied and at least one active node exists in the evolutionary history. Otherwise, the simulation ends.

This process simultaneously simulates, forward in time, evolutionary history and protein evolution, with protein evolution influencing the evolutionary history through selection from the folding stability. Indeed, selection can vary among protein variants at their corresponding nodes of the evolutionary history. The process produces a forward in time birth-death phylogenetic history that encompasses nodes that reached the ending time, internal nodes, and nodes that were extinct at some time, along with the protein variant associated with each node.

- The method includes several optional capabilities listed in **Supplementary file 1A** and in the software documentation, with some summarized below.
- The user can fix the birth and death rates along the evolutionary history or specify that they are based on the fitness of the corresponding protein variant as described before. For the latter, the fitness can be based on the folding stability of the analyzed protein variant or on the similarity in folding stability between the analyzed and real protein variants. In addition, the global birth-death rate can be constant or vary among lineages depending on the specified birth-death model.
- The birth-death simulation can finish when any of the following criteria is met: (a) A specified sample size, including extinction nodes derived from death events. (b) A specified sample size, excluding extinction nodes. (c) A specified evolutionary time, measured from the root to a tip node.
- Extinction nodes can either be preserved or removed from the evolutionary history.
- Evaluation of the simulated birth-death evolutionary history in terms of tree balance using the Colless index (**Colless and Wiley, 1982**; **Lemant et al., 2022**).
- Possible variation of the site-specific substitution rate according to user specifications.
- Customizable substitution model featuring site-specific exchangeability matrices (relative rates of change among amino acids and their frequencies at the equilibrium).

- Implementation of several SCS models and a variety of empirical substitution models of protein evolution. The implemented SCS models were evaluated in our previous work (*Arenas et al., 2013*), and the implemented empirical substitution models were properly identified using simulated data with *ProtTest3* (*Darriba et al., 2011*).

The implemented SCS models consider molecular energy functions based on amino acid contact matrices and configurational entropies per residue in unfolded and misfolded proteins (*Arenas et al., 2013*; *Bastolla et al., 2007*; *Mendez et al., 2010*). These models incorporate both positive and negative design strategies. In particular, the evaluation of the target protein structure while taking into account a database of residue contacts from alternative protein structures in the PDB, thus considering background genetic information that helps reduce prediction biases (*Minning et al., 2013*). Technical details about these SCS models are presented in our previous study (*Arenas et al., 2013*). Next, SCS models outperformed models that ignore structural evolutionary constraints in terms of phylogenetic likelihood, among other properties (*Arenas et al., 2013*; *Arenas et al., 2016a*; *Bordner and Mittelmann, 2014*). The method also implements common empirical substitution models of protein evolution (i.e. *Blosum62*, *CpRev*, *Dayhoff*, *DayhoffDCMUT*, *FLU*, *HIVb*, *HIVw*, *JTT*, *JonesDCMUT*, *LG*, *Mtart*, *Mtmam*, *Mtrev24*, *RtRev*, *VT*, and *WAG*; *Supplementary file 1A*) and the user can specify any particular exchangeability matrix for all sites or for each site, allowing for heterogeneity of the substitution process among sites (details in *Supplementary file 1A* and in the software documentation). In addition, the framework implements heterogeneous substitution rates among sites by the traditional Gamma distribution (+G; *Yang, 1996*) and proportion of invariable sites (+I; *Fitch and Margoliash, 1967*), and also the user can directly alter the substitution rate at each site for any empirical or SCS model (Table S1). Regarding the evolutionary history, in addition to the birth-death process presented before, the user can specify a particular phylogenetic tree or simulate a coalescent evolutionary history (*Hudson, 1983*; *Kingman, 1982*; *Supplementary file 1A*). In this regard, we maintained the capabilities of the previous version, including the coalescent with recombination (*Hudson, 1983*) which can be homogeneous or heterogeneous along the sequence according to *Wiuf and Posada, 2003*, variable population size over time (growth rate or demographic periods), several migration models that is island (*Hudson, 1998*), stepping-stone (*Kimura and Weiss, 1964*), and continent-island *Wright, 1931* with temporal variation of migration rates and convergence of demes or subpopulations, simulation of haploid or diploid data, and longitudinal sampling (*Navascués et al., 2010*; details in *Supplementary file 1A* and in the software documentation). The framework outputs a simulated multiple sequence alignment with the protein sequences of the internal and tip nodes, as well as their folding stabilities and the evolutionary history, among other information (*Supplementary file 1A* and software documentation).

## Study data for evaluating the method for forecasting protein evolution

We evaluated the accuracy of the method for forecasting protein evolution using viral proteins sampled over time (longitudinal sampling). Specifically, we used protein sequences from previous experiments

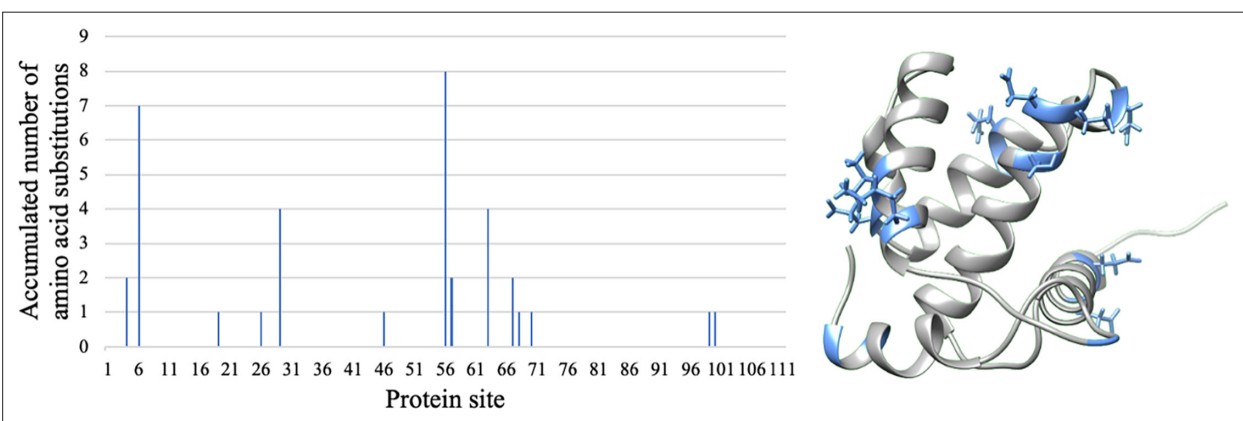

**Figure 2.** Distribution of amino acid substitutions observed along the HIV-1 MA sequences at time *T31*. Left: Distribution of the observed amino acid substitutions along the HIV-1 matrix (MA) protein sequences at time *T31*. Right: Distribution of the indicated amino acid substitutions (shown in blue) along the protein structure.

of virus evolution monitored over time, which contain consensus molecular data (avoiding rare variants) that belong to different evolutionary time points.

- The matrix (MA) protein of HIV-1, with data obtained from an in vitro cell culture experiment where samples were collected at different times (*Arenas et al., 2016b*). This data included 21 MA protein consensus sequences collected at times (population passages, *T*) *T1* (initial time and includes an initial sequence) and *T31* (time after 31 passages and includes 20 sequences), which accumulated 48 amino acid substitutions (sequence identity 0.973; *Figure 2*).
- The main protease (Mpro) and papain-like protease (PLpro) proteins of SARS-CoV-2. For each protein, the data includes the first sequenced variant (Wuhan) and a sequence built with all the substitutions observed in a dataset of 384 genomes of the Omicron variant of concern collected from the GISAID database. The resulting Mpro and PLpro sequences presented 10 and 22 substitutions (sequence identity 0.967 and 0.930), respectively.
- The non-structural protein 1 (NS1) of the influenza virus. We retrieved the NS1 sequences from the years 2005, 2015 and 2020 from the Influenza Virus Resource (*Bao et al., 2008*). Next, we obtained the consensus sequence of the 2005 dataset as initial time point (*T1*), and the consensus sequences from the 2015 (*T2*) and 2020 (*T3*) datasets as subsequent time points. The resulting consensus sequences for 2015 and 2020 showed 40 and 37 substitutions (sequence identity 0.802 and 0.817), respectively.
- The protease (PR) of HIV-1 with data sampled from patients monitored over time, from 2008 to 2017, available from the Specialized Assistance Services in Sexually Transmissible Diseases and HIV/AIDS in Brazil (*Ferreiro et al., 2022*; *Santos-Pereira et al., 2021*; *Souto et al., 2021*). These evolutionary scenarios are complex due to the diverse antiretroviral therapies administered to the patients (*Supplementary file 1B*), which could vary during the studied time periods and that could promote the fixation of specific mutations (i.e. associated with resistance; *Ferreiro et al., 2022*; *Santos-Pereira et al., 2021*; *Souto et al., 2021*). For each viral population (patient), the data included a consensus sequence for each of the four or five samples collected at different time points. These consensus sequences exhibited between one and 22 amino acid substitutions with respect to the consensus sequence of the corresponding first sample (*Supplementary file 1B*).

We identified a representative protein structure for each dataset, which was used to predict the folding stability and to inform the SCS model. In particular, we obtained the protein structures of the sequences at the initial time (*T1*) from the PDB (*Supplementary file 1C*). For the case of the HIV-1 protease, we obtained the protein structure through homology modeling. We used *SWISS-MODEL* (*Arnold et al., 2006*) to identify the best-fitting templates of PDB structures (*Supplementary file 1C*). Next, we predicted the protein structures by homology modeling with *Modeller* (*Sali and Blundell, 1993*) *using the protein sequences and corresponding best-fitting structural templates.*

## Forward in time prediction of viral protein variants

The evaluation was performed with the previously presented real data, which includes a real protein variant present at the initial time (*T1*) and subsequent protein variants present at a later time (*Tn*). We applied the method for forecasting protein evolution to predict the most likely protein variants at *Tn* derived from the real protein variant observed at the initial time (*T1*). The prediction error was determined by measuring the distance between the real protein variants and the predicted variants corresponding to time *Tn*.

Thus, we assigned to the initial node (*T1*) the corresponding real protein variant (including its sequence and structure), and its evolution was simulated forward in time until it reached the number of substitutions observed in the real data at time *Tn*. Thus, we considered the number of observed substitutions as a measure of evolutionary time to allow proper comparisons between real and predicted variants. We simulated 100 alignments of protein sequences, each containing the same number of sequences as the real data at *Tn*. This included 20 HIV-1 MA protein sequences, a consensus sequence for the SARS-CoV-2 Mpro, a consensus sequence for the SARS-CoV-2 PLpro, a consensus sequence of the influenza NS1 protein for each time point, and a consensus sequence of HIV-1 PR for each viral population. To avoid rare variants, each sequence of the simulated multiple sequence alignments was obtained as the consensus of 100 linked simulated sequences.

To investigate the effect of selection on the predictions, we compared the accuracy of forecasting protein evolution when selection from the protein structure is considered and when it is ignored (neutral evolution). If selection is considered, as previously presented, the probability of birth and

death events was based on the fitness of the protein variants, and protein evolution was modeled using an SCS model (*Arenas et al., 2013*). In the case of neutral evolution, all protein variants equally fit and are allowed. Since variants are observed, we allowed birth events. However, it assumed the absence of death events as no information independent of fitness is available to support their inclusion, thereby avoiding the imposition of arbitrary death events based on an arbitrary death rate. Also, to model neutral evolution, we used an exchangeability matrix with the same relative rates of change to all amino acid pairs.

## Accuracy of the predicted protein variants

We assessed the accuracy of the method for forecasting protein evolution by comparing the predicted and real protein variants present at time $Tn$, both at the protein sequence and structure levels.

For data containing multiple sequences at time $Tn$ (i.e. HIV-1 MA dataset), we calculated the Kullback-Leibler (KL) divergence, which provides a distance between two multiple sequence alignments (the real and predicted data) based on the distribution of amino acid frequencies along the sequences (Equation III, where factors $P$ and $Q$ denote the distribution of amino acid frequencies in the real and predicted protein sequences at time $Tn$, respectively, $i$ refers to protein site) (*Kullback and Leibler, 1951*). This distance was only calculated for data with a set of sequences at time $Tn$ (HIV-1 MA) because a single sequence does not provide site-specific variability.

$$KL(P \parallel Q) = \sum_i P_i \times \log(\frac{P_i}{Q_i})$$

(3)

We also compared the real and predicted evolutionary trajectories of protein variants using the Grantham distance, which measures the differences between amino acids based on their physicochemical properties (*Grantham, 1974*). In particular, for both real and predicted protein variants, we calculated the Grantham distance at each protein site that differs between the two datasets, considering its evolution from $T1$ to the subsequent multiple sequence alignment at $Tn$. We examined sites that varied over time, thus the general site-specific Grantham distance $Gi$ was calculated as the frequency of each amino acid $f$ at site $i$ multiplied by the specific Grantham distance between amino acid $m$ at time $Tn$ and amino acid $n$ at time $T1$, normalized with the largest Grantham distance $G_{max}$ to obtain values between 0 and 1 (Equation IV). Next, to compare the real and predicted data, we calculated the site-specific difference of Grantham distance $Gb_i$ between the real $P$ and predicted $Q$ protein variants (Equation V),

$$G_i = \sum_{m=1}^{20} f_m \times \frac{G_{(m,n)}}{G_{max}}$$

(4)

$$Gb_i = \mid G_{P,i} - G_{Q,i} \mid$$

(5)

In addition, we obtained and compared the protein folding stability (ΔG) of the predicted and real protein variants observed at time $Tn$, using their corresponding protein structures, with *DeltaGREM* (*Arenas et al., 2016a*; *Minning et al., 2013*).

## Results

### Implementation of the forecasting protein evolution method

We extended the previous version of our framework *ProteinEvolver* (*Arenas et al., 2013*), maintaining its previous capabilities (i.e. simulation of protein evolution upon user-specified phylogenetic trees and upon phylogenetic trees simulated with the coalescent with or without recombination, migration, demographics and longitudinal sampling, empirical and SCS models, among others; *Supplementary file 1A*), by adding, among others (*Supplementary file 1A*), the forward in time modeling of protein evolution that combines a birth-death process based on the fitness of every protein variant (folding stability) at each node to determine its birth and death rates, as well as SCS models of protein evolution. The framework *ProteinEvolver2* is written in C and distributed with a detailed documentation and a variety of illustrative practical examples. The framework is freely available from https://github.com/MiguelArenas/proteinevolver (*Arenas, 2025*).

**Table 1.** Comparison of real and predicted sequences of the HIV-1 MA protein considering predictions based on the SCS and neutral models.

For the data simulated under the SCS [including birth-death models with constant (SCS) and variable global birth-death rate among lineages (GlobalBDvar)] and neutral models, the table shows the Grantham distance between the amino acids that changed during the real and predicted evolutionary trajectories and the Kullback-Leibler (KL) divergence between the real and predicted multiple sequence alignments. Next, it shows the folding stability (ΔG) of the real protein variants at times *T1* and *T31* and the folding stability of the predicted protein variants at time *T31*. The error corresponds to the 95% confidence interval from the mean (100 samples) of predictions of folding stability.

| | Grantham distance | KL divergence | ΔG of the real variant at *T1* (kcal/mol) | ΔG of the real variants at *T31* (kcal/mol) | ΔG of the predicted variants at *T31* (kcal/mol) | ΔΔG (kcal/mol) at *T31* (predicted – real variants) |
|---|---|---|---|---|---|---|
| SCS model | 5% | 6% | –9.72 | –10.34±0.14 | –9.96±0.02 | 0.38 |
| SCS GlobalBDvar model | 5% | 6% | –9.72 | –10.34±0.14 | –10.03±0.03 | 0.31 |
| Neutral model | 5% | 6% | –9.72 | –10.34±0.14 | –9.21±0.07 | 1.14 |

### Evaluation of predictions of HIV-1 MA evolution

Regarding the evolution of the HIV-1 MA protein, the Grantham distance and the KL divergence between the real variants at time *Tn* and the corresponding predicted variants were low (around 5% and 6%, respectively; *Table 1*), and they did not differ comparing predictions that consider selection on the folding stability (including birth-death models with constant and variable global birth-death rate among lineages) and predictions that ignore it (*Table 1*). On the other hand, we found that the folding stability of the protein variants predicted considering selection on the folding stability (again, including birth-death models with constant or variable global birth-death rate among lineages) was closer to the folding stability of the real protein variants than that of the protein variants predicted under neutral evolution (*Table 1*). In particular, the protein variants predicted ignoring selection were less stable than those predicted considering selection and also less stable than the real protein variants.

### Evaluation of predictions of SARS-CoV-2 Mpro and PLpro evolution

The analyses of the SARS-CoV-2 Mpro and PLpro data showed Grantham distances between the real and predicted sequences around 25% and 36%, respectively (*Figure 3A*). Again, this distance was similar when comparing predictions based on models that consider selection on the protein folding stability (including birth-death models with constant or variable global birth-death rate among lineages) and a model of neutral evolution. Regarding comparisons based on the protein folding stability, we found again that the models that consider selection from the folding stability produce variants closer to the stability of the real protein variants than the model that ignores selection (*Figure 3B*). Indeed, protein variants derived from the models that consider selection were more stable than those derived from the model of neutral evolution. Next, we did not find statistically significant differences in sequence similarity or folding stability between variants predicted under birth-death models with constant or variable global birth-death rate among lineages (*Figure 3*).

### Evaluation of predictions of influenza NS1 protein evolution

The evolutionary predictions for the influenza NS1 protein varied depending on the model used. Specifically, at the sequence level and for the two prediction time points studied, Grantham distances between the real and predicted protein sequences were around 23.5% for the models that incorporated structural evolutionary constraints, compared to about 25.5% for the neutral model (*Figure 4A*). These differences became more pronounced when examining predictions based on protein folding stability. For both time points, models that included selection consistently generated protein variants with stability more similar to that of the observed variants than those predicted by the neutral model (*Figure 4B*). Indeed, sequences predicted by the model that accounts for selection were generally more stable than those predicted under neutral evolution. Again, we found no statistically significant differences in sequence similarity or folding stability between variants predicted under birth-death models with constant or variable global birth-death rate among lineages (*Figure 4*).

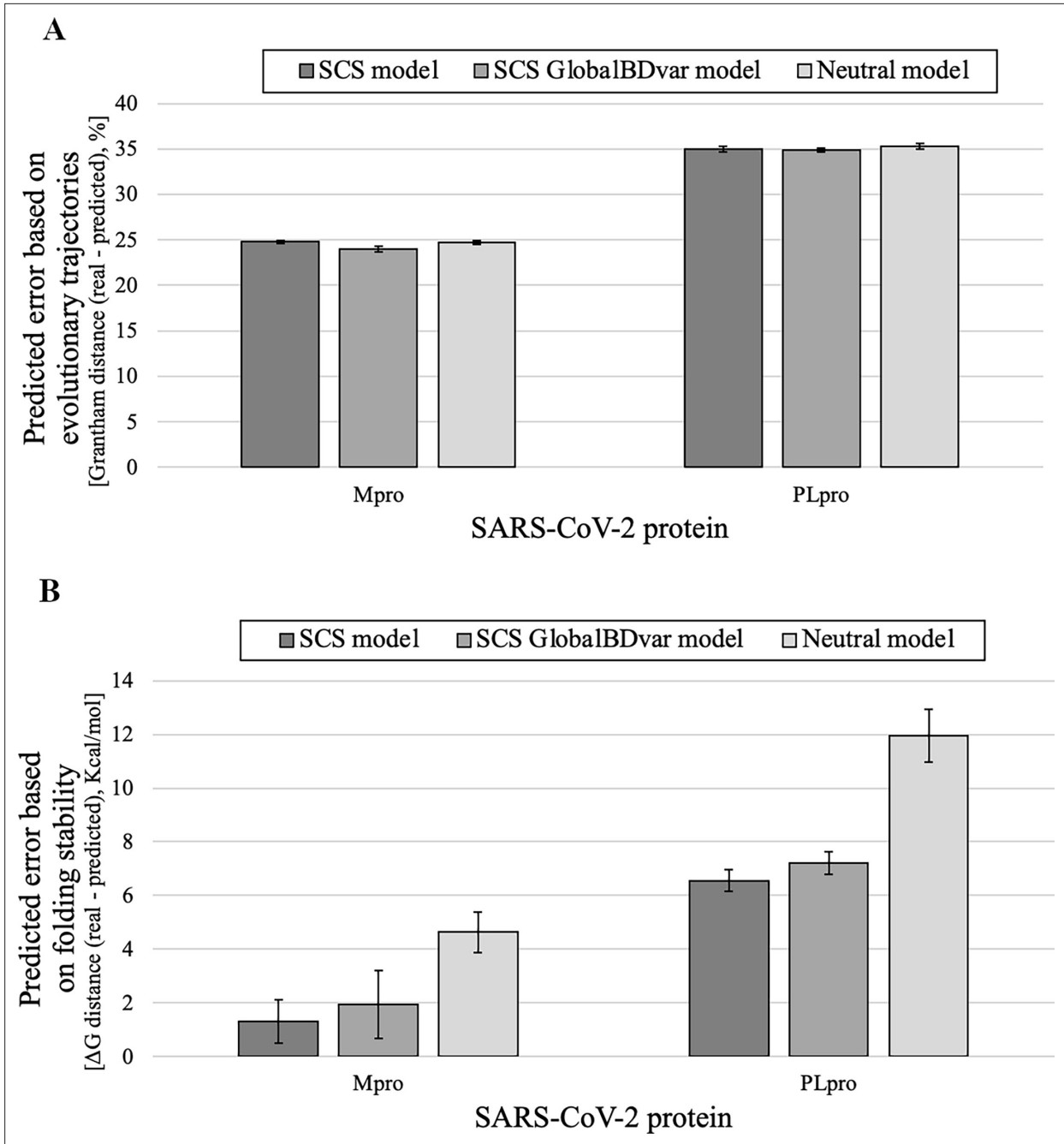

**Figure 3.** Prediction error of SARS-CoV-2 Mpro and PLpro evolution under SCS and neutral models regarding physicochemical properties of the amino acid changes accumulated during the evolutionary trajectories and protein folding stability. Predictions based on data simulated under the SCS [including birth-death models with constant (SCS) and variable global birth-death rate among lineages (GlobalBDvar)] and neutral models. (**A**) Grantham distance calculated from the amino acid changes that occurred during the real and predicted evolutionary trajectories based on SCS and neutral models of protein evolution. (**B**) Variation of protein folding stability (ΔΔG) between real and predicted protein variants based on SCS and neutral models of protein evolution. Notice that positive ΔΔG indicates that the real protein variants are more stable than the predicted protein variants and *vice versa*. Error bars correspond to the 95% confidence interval of the mean of prediction error from 100 multiple sequence alignments simulated for the corresponding population and time.

## Evaluation of predictions of HIV-1 PR evolution

In general, the Grantham distance, which compared the evolutionary trajectories of the real and predicted protein variants from time *T1* to later times, varied among viral populations (patients; *Figure 5A*). However, for the majority of these populations, the distance remained below 30% and

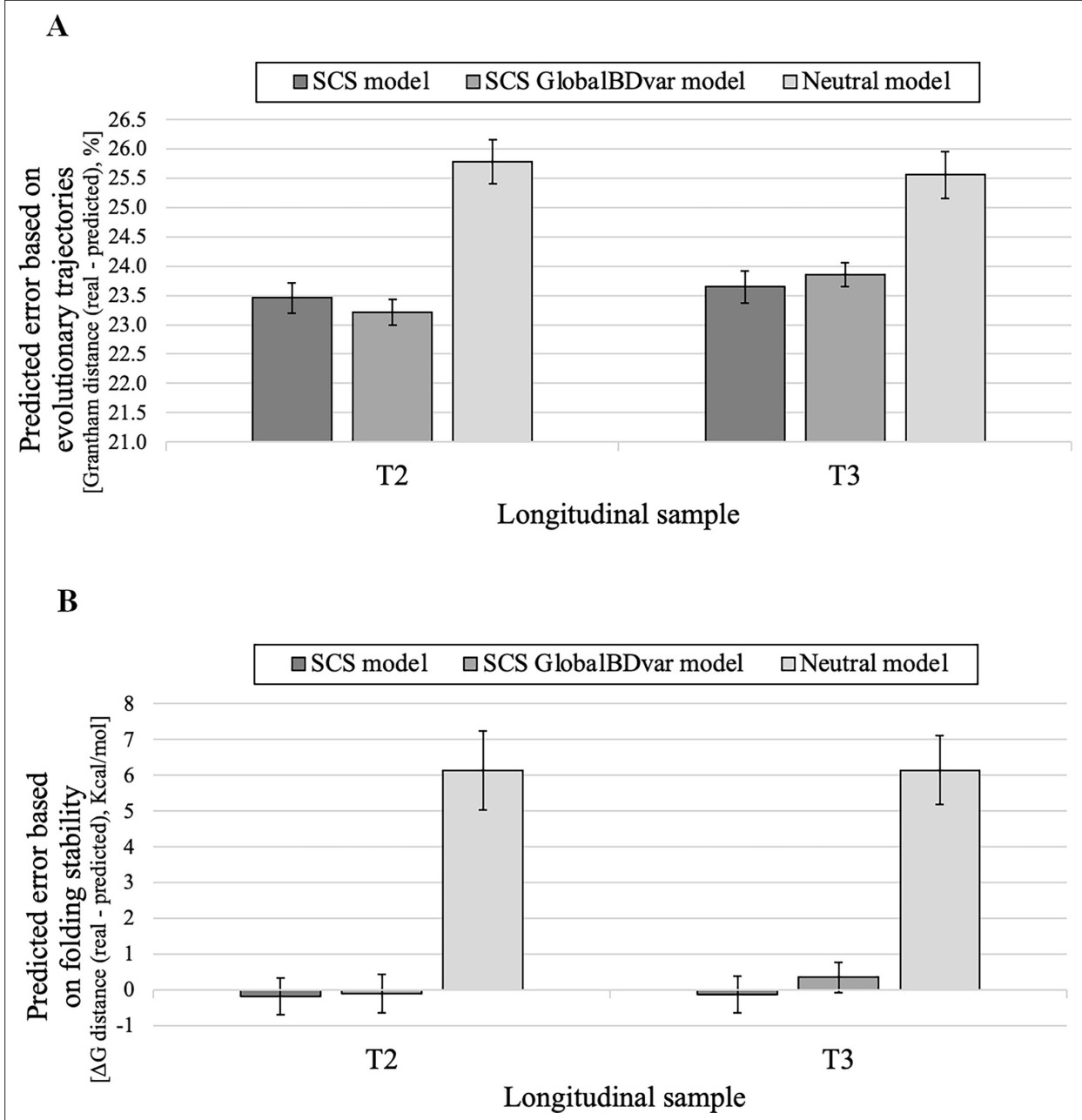

**Figure 4.** Prediction error of influenza NS1 protein evolution under SCS and neutral models regarding physicochemical properties of the amino acid changes accumulated during the evolutionary trajectories and protein folding stability. Predictions were performed for two time points (longitudinal samples T2 and T3). Predictions based on data simulated under the SCS [including birth-death models with constant (SCS) and variable global birth-death rate among lineages (GlobalBDvar)] and neutral models. (**A**) Grantham distance calculated from the amino acid changes that occurred during the real and predicted evolutionary trajectories based on SCS and neutral models of protein evolution. (**B**) Variation of protein folding stability (ΔΔG) between real and predicted protein variants based on SCS and neutral models of protein evolution. Notice that positive ΔΔG indicates that the real protein variants are more stable than the predicted protein variants and vice versa. Error bars correspond to the 95% confidence interval of the mean of prediction error from 100 multiple sequence alignments simulated for the corresponding population and time.

exhibited minor fluctuations over time. One particular population exhibited a notable trend, with the distance increasing from 10% to nearly 60% over time. Considering that the length of the evolutionary trajectories of the protein can differ among the studied populations, we explored whether the accumulated number of amino acid substitutions could affect the accuracy of the predictions, and we found that the number of substitutions varied among populations and this variability did not

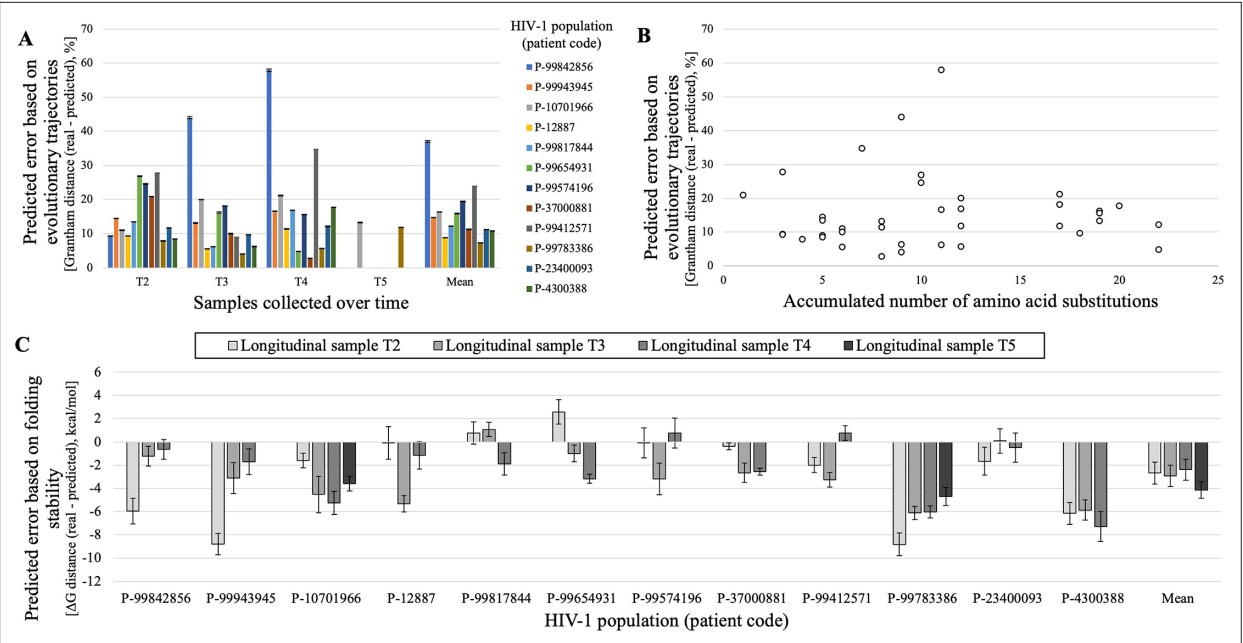

**Figure 5.** Prediction error of HIV-1 PR evolution at diverse populations regarding physicochemical properties of the amino acid changes accumulated during the evolutionary trajectories and protein folding stability. (**A**) For each viral population (patient, represented with a particular color) and time, Grantham distance calculated from the amino acid changes that occurred during the real and predicted evolutionary trajectories. For each population, the mean of distances obtained over time is shown on the right. (**B**) Relationship between Grantham distances and accumulated number of substitution events ($R^2$=0.0001, which indicates a lack of correlation between these parameters). (**C**) Variation of protein folding stability ($\Delta\Delta G$) between real and predicted protein variants at each viral population and time. For each population, the mean of distances obtained over time is shown on the right. Notice that positive $\Delta\Delta G$ indicates that the real protein variants are more stable than the predicted protein variants and *vice versa*. Error bars correspond to the 95% confidence interval of the mean of prediction error from 100 multiple sequence alignments simulated for the corresponding viral population and time.

correlate with the Grantham distance between the real and predicted data ($R^2$=0.0001, *Figure 5B*). In general, the folding stability of the predicted protein variants was similar or slightly less stable than that of the real protein variants [with a difference ranging from 0 to 9 kcal/mol and a mean of 3.1±0.9 (95% CI) kcal/mol; *Figure 5C*]. Indeed, the folding stability exhibited small fluctuations, increasing and decreasing, over time.

## Discussion

While reconstructing evolutionary histories and ancestral sequences, among other inferences about the past, was popular in the field, predictions about evolution toward the future were traditionally ignored due to potential high prediction errors. However, in the last decade, forecasting evolution has gained attention because of its variety of applications and advancements in models of evolution (see the reviews *Lässig et al., 2017*; *Wortel et al., 2023*). Several studies showed that forecasting evolution can be feasible in some systems, including virus evolution (*Luksza and Lässig, 2014*; *Thadani et al., 2023*). Here, we also investigated forecasting evolution in viruses but at the molecular level, considering the relevance of the substitution process to produce molecular variants that affect the viral fitness (*Arenas, 2015a*; *Arenas et al., 2016b*; *Bloom and Neher, 2023*; *Poon et al., 2007*; *Watabe and Kishino, 2010*). We believe that forecasting genome evolution remains challenging due to the complexity of its evolutionary processes [i.e. including epistatic interactions, chromosomal rearrangements, and heterogeneous substitution patterns among genomic regions (*Arbiza et al., 2011*), among others] that complicate the parameterization and prediction of accurate fitness landscapes. However, we believe that it could be more feasible in structural proteins because of their relatively simpler evolutionary processes that usually include selection on folding stability (*Bloom et al., 2006*; *Goldstein, 2011*).

To make evolutionary predictions over time, either toward the past or toward the future, considering a substitution model of molecular evolution can be convenient. Actually, a variety of traditional studies showed that the substitution model influences the phylogenetic likelihood generated by probabilistic approaches used for evolutionary inference (*Lemmon and Moriarty, 2004*; *Yang et al., 1994*; *Zhang and Nei, 1997*). To study protein evolution, empirical substitution models of protein evolution are routinely used because they allow rapid calculations based on the assumption of site-independent evolution and typically apply the same exchangeability matrix for all the protein sites, which is highly unrealistic (*Echave et al., 2016*). Besides, a small set of empirical substitution models was developed for modeling the evolution of the diverse range of viral proteins (*Dang et al., 2010*; *Del Amparo and Arenas, 2022b*; *Dimmic et al., 2002*; *Le and Vinh, 2020*; *Nickle et al., 2007*). As an alternative, some substitution models that consider evolutionary constraints on the protein structure incorporated site-dependent evolution, which allows a more accurate modeling of protein evolution compared to the empirical substitution models (*Arenas et al., 2013*; *Arenas et al., 2016a*; *Bordner and Mittelmann, 2014*; *Ferreiro et al., 2024a*; *Parisi and Echave, 2005*).

According to previous methods for forecasting evolution based on computer simulations (*Lässig and Łuksza, 2014*; *Neher et al., 2014*), we also adopted the birth-death population genetics process to allow a forward-in-time evolutionary process where the birth and death rates can differ among nodes (*Stadler, 2010*; *Stadler, 2011*). Traditional population genetics methods to simulate the evolution of molecular data implement a two-step simulation process, where, first, the evolutionary history is simulated [i.e. with the coalescent theory or a forward-in-time simulation approach (*Arenas, 2012*; *Hoban et al., 2012*), often assuming neutral evolution] and, in a subsequent step, molecular evolution is simulated upon the previously obtained evolutionary history (*Yang, 2006*). However, this methodology is unrealistic because the fitness of the data can affect its evolutionary history (i.e., a variant with high fitness is likely to have more descendants than a variant with low fitness). Thus, we designed and implemented a method for forecasting protein evolution that integrates a birth-death population genetics process (including the modeling of constant and variable global birth-death rate among lineages) with a SCS model of protein evolution, where the folding stability of each protein variant is evaluated to predict its future trajectory in terms of both evolutionary history and molecular evolution. We implemented this forward-in-time simulation of protein evolution in a new version of our previous framework *ProteinEvolver* (*Arenas et al., 2013*), while maintaining its previous capabilities and extending some of them (i.e. incorporation of site-specific exchangeability matrices and additional substitution models of protein evolution, among others; see *Supplementary file 1A* and software documentation). As any other method for forecasting evolution, the present method ignores possible environmental shifts that are inherently unpredictable and that could affect the accuracy of the predictions. Next, we evaluated the forward-in-time evolutionary predictions with real data of HIV-1, SARS-CoV-2, and influenza virus proteins. We determined the prediction errors between the real and the predicted protein variants by examining dissimilarity in evolutionary trajectory (Grantham distance based on physicochemical properties among the differing amino acids during the evolutionary trajectories), sequence divergence (distribution of amino acid frequencies among sites using the KL divergence), and protein structure (protein folding stability). Additionally, we analyzed the influence of accounting for selection, based on the protein folding stability, on the predictions. We also evaluated birth-death models incorporating either constant or variable global birth-death rates among lineages. Notably, in the variable-rate model, fitness can influence both reproductive success and the rate of molecular evolution.

In general, the sequence and evolutionary trajectory dissimilarities between the real and predicted protein variants were relatively small, with some variations among the study proteins. For the HIV-1 MA protein, the SARS-CoV-2 Mpro and PLpro, and influenza NS1 protein, the sequence dissimilarity was below 10%, 26%, 36%, and 26%, respectively. The low prediction errors for the HIV-1 MA protein were expected because this dataset was derived from an in vitro cell culture experiment that is not influenced by a variety of external factors that could affect the predictions in in vivo experiments. As a rough reference, in traditional ancestral sequence reconstruction (ASR) of protein data with high sequence identity, the error was approximately 2% (*Arenas and Posada, 2010*). Notice that ASR methods can exhibit low prediction errors due to their statistical evaluation based on a set of original sequences, rather than relying on a single original sequence used in the case of forecasting evolution. The contrasting scenario regarding evolutionary complexity was the HIV-1 PR data, which

was sampled from populations (patients) undergoing various antiretroviral treatments (*Ferreiro et al., 2022*). There, the sequence dissimilarity varied among viral populations, although most were below 30%. A viral population exhibited higher dissimilarities (near 60%) that we believe could be caused by molecular adaptations promoted by the therapies, although this needs formal investigation.

An unexpected result was that the model of neutral evolution produced sequence dissimilarities between the real and predicted protein variants that were quite similar to those obtained with the SCS model (*Table 1* and *Figures 3 and 4*). A few studies indicated that the substitution model has negligible effects on the reconstructed phylogenetic trees (*Abadi et al., 2019*; *Spielman, 2020*). Subsequent studies found that the influence of the substitution model on phylogenetic reconstructions is dependent on the diversity of the data, where data with high diversity is more sensitive to the applied substitution model due to containing more evolutionary information to be modeled (*Del Amparo and Arenas, 2023*). Considering that the studied data present overall low diversity [i.e. sequence identity of 0.973, 0.967, 0.930, 0.802, and 0.817 for the HIV MA data, SARS-CoV-2 Mpro data, SARS-CoV-2 PLpro data, and influenza NS1 protein data (for each prediction time point, T2 and T3), respectively; it is important to note that, in general, longitudinal data derived from monitored evolutionary processes usually show a low diversity because they involve relatively short evolutionary histories, among other factors.], we believe that the influence of the applied substitution models on the prediction of the sequences was small because of the small number of modeled substitution events, as found in phylogenetic reconstructions (; *Del Amparo and Arenas, 2023*). Actually, in the case of the influenza NS1 protein dataset, which had the highest sequence diversity among the study datasets, the sequences predicted under the SCS models were more similar to the real sequences than those derived from predictions under neutral evolution. Overall, the prediction accuracy varied among the studied evolutionary scenarios; as expected, it was lower in the more complex scenarios. Indeed, datasets with higher sequence diversity contain more evolutionary signals, which can improve prediction quality.

We also evaluated the prediction error between the real and predicted protein variants regarding their folding stability, again comparing the predictions made under a model that considers structural constraints and a model of neutral evolution. In general, the protein variants predicted under the SCS model presented a folding stability close to the folding stability of the respective real protein variants, with differences below 1, 2, 7, and 1 kcal/mol for the HIV MA data, SARS-CoV-2 Mpro data, SARS-CoV-2 PLpro data, and influenza NS1 protein data, respectively. The higher differences (around 9 kcal/mol) were again observed for the HIV-1 PR data. In contrast to the prediction error based on sequence dissimilarity, the prediction error based on folding stability varies between predictions obtained under the SCS model and those obtained under the neutral model. In the studied evolutionary scenarios, the protein variants predicted under the neutral model were less stable and farther from the stability of the real protein variants compared to those predicted under the SCS model. These results were expected because, under SCS models, protein stability can be modeled with greater accuracy than sequence similarity due to selection for maintaining stability in the protein structure despite amino acid changes (*Arenas and Bastolla, 2020*; *Illergård et al., 2009*; *Pascual-García et al., 2010*). Indeed, previous studies showed that models that ignore structural constraints often produce proteins with unrealistic folding instability (*Arenas and Bastolla, 2020*; *Del Amparo et al., 2023*), which suggests that accounting for protein folding stability in the modeling of protein evolution is recommended for predicting protein variants with appropriate structural properties.

Therefore, we found a good accuracy in predicting the real folding stability of forecasted protein variants, while predicting the exact sequences was more challenging, which was not surprising considering previous studies. In particular, inferring specific sequences is considerably challenging even for ancestral molecular reconstruction (*Arenas and Bastolla, 2020*; *Arenas et al., 2017*). Indeed, observed sequence diversity is much greater than observed structural diversity (*Illergård et al., 2009*; *Pascual-García et al., 2010*), and substitutions between amino acids with similar physicochemical properties can yield modeled protein variants with more accurate folding stability, even when the exact amino acid sequences differ. Further work is demanded in the field of substitution models of molecular evolution. We also found that datasets with relatively high sequence diversity can improve the accuracy of the predictions due to containing more evolutionary information. Apart from that, forecasting the folding stability of future real proteins is an important advance in forecasting protein evolution, given the essential role of folding stability in protein function (*Bloom et al., 2006*; *Scheiblhofer et al., 2017*) and its variety of applications.

Variation in the global birth-death rate among lineages showed minor effects on prediction accuracy, suggesting a limited role in protein evolution modeling. Molecular evolution parameters, particularly the substitution model, appear to be more critical in this regard.

In the context of protein evolution, substitution models are a critical factor (*Arenas et al., 2017*; *Bordner and Mittelmann, 2014*; *Echave et al., 2016*; *Echave and Wilke, 2017*; *Liberles et al., 2012*; *Wilke, 2012*), and the presented combination with a birth-death model constitutes a first approximation upon which next studies can build to better understand this evolutionary system. Next, the present method assumes that the protein structure is maintained over the studied evolutionary time, which can be generally reasonable for short timescales where the structure is conserved (*Illergård et al., 2009*; *Pascual-García et al., 2010*). Over longer evolutionary timescales, structural changes may occur and, in such cases, modeling the evolution of the protein structure would be necessary. To our knowledge, modeling the evolution of the protein structure remains a challenging task that requires substantial methodological developments. Recent advances in artificial intelligence, particularly in protein structure prediction from sequence (*Abramson et al., 2024*; *Jumper et al., 2021*), may offer promising tools for addressing this challenge. However, we believe that evaluating such approaches in the context of structural evolution would be difficult, especially given the limited availability of real data with known evolutionary trajectories involving structural change. In any case, this is probably an important direction for future research.

We present a method to simulate forward-in-time protein evolution accounting for evolutionary constraints from the protein structure and a birth-death population process, and where the evolutionary history is influenced by the protein evolution and vice versa. The method is implemented in the computer framework *ProteinEvolver2*, which is freely distributed with several practical examples and detailed documentation. We believe that implementing methods into freely available phylogenetic frameworks is important to facilitate practical applications, as well as future improvements and evaluations. We applied the method to forecast protein evolution in some viral proteins. We found that the method provides acceptable approximations to the real evolution, especially in terms of protein folding stability, suggesting that combining structural constraints with birth-death population processes in the modeling of protein evolution is convenient. Still, to advance in methods for forecasting protein evolution, we believe that further efforts should be made in the field to improve the modeling of protein evolution, such as the incorporation of site-dependent evolutionary constraints from the protein activity.

## Acknowledgements

This work was supported by the Project PID2023-151032NB-C22 funded by MCIU/AEI/10.13039/501100011033 and by FEDER, UE. DF was funded by a fellowship from Xunta de Galicia [ED481A-2020/192]. We thank the "Centro de Supercomputación de Galicia (CESGA)" for the computer resources. The funders had no role in study design, data collection and analysis, decision to publish, or preparation of the manuscript.

## Additional information

### Funding

| Funder | Grant reference number | Author |
| --- | --- | --- |
| Ministerio de Ciencia, Innovación y Universidades | PID2023-151032NB-C22 | Luis Daniel González-Vázquez Miguel Arenas |
| Agencia Estatal de Investigación | PID2023-151032NB-C22 | Luis Daniel González-Vázquez Miguel Arenas |
| Federación Española de Enfermedades Raras | | Luis Daniel González-Vázquez Miguel Arenas |
| Xunta de Galicia | ED481A-2020/192 | David Ferreiro |

| Funder | Grant reference number | Author |
|--------|------------------------|--------|

The funders had no role in study design, data collection and interpretation, or the decision to submit the work for publication.

## Author contributions

David Ferreiro, Resources, Data curation, Software, Formal analysis, Supervision, Validation, Investigation, Visualization, Methodology, Writing – original draft, Writing – review and editing; Luis Daniel González-Vázquez, Resources, Data curation, Formal analysis, Investigation, Visualization, Methodology, Writing – review and editing; Ana Prado-Comesaña, Resources, Data curation, Formal analysis, Investigation, Visualization, Writing – review and editing; Miguel Arenas, Conceptualization, Resources, Software, Funding acquisition, Investigation, Visualization, Methodology, Writing – original draft, Project administration, Writing – review and editing

## Author ORCIDs

David Ferreiro ⓘ https://orcid.org/0000-0003-0757-7702
Luis Daniel González-Vázquez ⓘ https://orcid.org/0000-0002-6973-6660
Miguel Arenas ⓘ https://orcid.org/0000-0002-0516-2717

Reviewer #1 (Public review): https://doi.org/10.7554/eLife.106365.3.sa1
Reviewer #2 (Public review): https://doi.org/10.7554/eLife.106365.3.sa2
Author response https://doi.org/10.7554/eLife.106365.3.sa3

# Additional files

## Supplementary files

Supplementary file 1. Supplementary tables A–C and references cited in the supplementary file.

MDAR checklist

## Data availability

The computer framework ProteinEvolver2 is freely available from https://github.com/MiguelArenas/proteinevolver (**Arenas, 2025**). The SARS-CoV-2 data is available from GISAID database with https://doi.org/10.55876/gis8.250206gt. The real and predicted protein variants are available from Zenodo repository at the URL https://doi.org/10.5281/zenodo.15548146.

The following dataset was generated:

| Author(s) | Year | Dataset title | Dataset URL | Database and Identifier |
|-----------|------|---------------|-------------|-------------------------|
| Ferreiro D, González-Vázquez LD, Prado-Comesaña A, Arenas M | 2025 | Forecasting protein evolution by combining birth-death population models with structurally constrained substitution models | https://doi.org/10.5281/zenodo.15548146 | Zenodo, 10.5281/zenodo.15548146 |

The following previously published dataset was used:

| Author(s) | Year | Dataset title | Dataset URL | Database and Identifier |
|-----------|------|---------------|-------------|-------------------------|
| GISAID | 2025 | SARS-CoV-2 protein sequences | https://doi.org/10.55876/gis8.250206gt | EpiCoV, 10.55876/gis8.250206gt |

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
