## [Editor Report · eLife Assessment]

This manuscript introduces a **useful** protein-stability-based fitness model for simulating protein evolution and unifying non-neutral models of molecular evolution with phylogenetic models. The model is applied to five viral proteins that are of structural and functional importance. While the general modelling approach is **solid**, and effectively preserves folding stability, the evidence for the model's predictive power remains limited, since it shows little improvement over neutral models in predicting protein evolution. The work should be of interest to researchers developing theoretical models of molecular evolution.

---

## [Referee Report · Reviewer #1 (Public review)]

Summary:

Ferreiro et al. present a method to simulate protein sequence evolution under a birth-death model where sequence evolution is guided by structural constraints on protein stability. The authors then use this model to explore the predictability of sequence evolution in several viral proteins. In principle, this work is of great interest to molecular evolution and phylodynamics, which has struggled to couple non-neutral models of sequence evolution to phylodynamic models like birth-death processes. Unfortunately, though, the model shows little improvement over neutral models in predicting protein sequence evolution, although it can predict protein stability better than models assuming neutral evolution. It appears that more work is needed to determine exactly what aspects of protein sequence evolution are predictable under such non-neutral phylogenetic models.

Major concerns:

(1) The authors have clarified the mapping between birth-death model parameters and fitness, but how fitness is modeled still appears somewhat problematic. The authors assume the death rate = 1 - birth rate. So a variant with a birth rate b = 1 would have a death rate d = 0 and so would be immortal and never die, which does not seem plausible. Also I'm not sure that this would "allow a constant global (birth-death) rate" as stated in line 172, as selection would still act to increase the population mean growth rate r = b - d. It seems more reasonable to assume that protein stability affects only either the birth or death rate and assume the other rate is constant, as in the Neher 2014 model.

(2) It is difficult to evaluate the predictive performance of protein sequence evolution. This is in part due to the fact that performance is compared in terms of percent divergence, which is difficult to compare across viral proteins and datasets. Some protein sequences would be expected to diverge more because they are evolving over longer time scales, under higher substitution rates or under weaker purifying selection. It might therefore help to normalize the divergence between predicted and observed sequences by the expected or empirically observed amount of divergence seen over the timescale of prediction.

(3) Predictability may also vary significantly across different sites in a protein. For example, mutations at many sites may have little impact on structural stability (in which case we would expect poor predictive performance) while even conservative changes at other sites may disrupt folding. I therefore feel that there remains much work to be done here in terms of figuring out where and when sequence evolution might be predictable under these types of models, and when sequence evolution might just be fundamentally unpredictable due to the high entropy of sequence space.

---

## [Referee Report · Reviewer #2 (Public review)]

In this study, the authors aim to forecast the evolution of viral proteins by simulating sequence changes under a constraint of folding stability. The central idea is that proteins must retain a certain level of structural stability (quantified by folding free energy, ΔG) to remain functional, and that this constraint can shape and restrict the space of viable evolutionary trajectories. The authors integrate a birth-death population model with a structurally constrained substitution (SCS) model and apply this simulation framework to several viral proteins from HIV-1, SARS-CoV-2, and influenza.

The motivation to incorporate biophysical constraints into evolutionary models is scientifically sound, and the general approach aligns with a growing interest in bridging molecular evolution and structural biology. The authors focus on proteins where immune pressure is limited and stability is likely to be a dominant constraint, which is conceptually appropriate. The method generates sequence variants that preserve folding stability, suggesting that stability-based filtering may capture certain evolutionary patterns.

However, the study does not substantiate its central claim of forecasting. The model does not predict future sequences with measurable accuracy, nor does it reproduce observed evolutionary paths. Validation is limited to endpoint comparisons in a few datasets. While KL divergence is used to compare amino acid distributions, this analysis is only applied to a single protein (HIV-1 MA), and there is no assessment of mutation-level predictive accuracy or quantification of how well simulated sequences recapitulate real evolutionary paths. No comparison is made to real intermediate variants available from extensive viral sequencing datasets which gather thousands of sequences with detailed collection date annotation (SARS-CoV-2, Influenza, RSV).

The selection of proteins is narrow and the rationale for including or excluding specific proteins is not clearly justified.

The analyzed datasets are also under-characterized: we are not given insight into how variable the sequences are or how surprising the simulated sequences might be relative to natural diversity. Furthermore, the use of consensus sequences to represent timepoints is problematic, particularly in the context of viral evolution, where divergent subclades often coexist - a consensus sequence may not accurately reflect the underlying population structure.

The fitness function used in the main simulations is based on absolute ΔG and rewards increased stability without testing whether real evolutionary trajectories tend to maintain, increase, or reduce folding stability over time for the particular systems (proteins) that are studied. While a variant of the model does attempt to center selection around empirical ΔG values, this more biologically plausible version is underutilized and not well validated.

Ultimately, the model constrains sequence evolution to stability-compatible trajectories but does not forecast which of these trajectories are likely to occur. It is better understood as a filter of biophysically plausible outcomes than as a predictive tool. The distinction between constraint-based plausibility and sequence-level forecasting should be made clearer. Despite these limitations, the work may be of interest to researchers developing simulation frameworks or exploring the role of protein stability in viral evolution, and it raises interesting questions about how biophysical constraints shape sequence space over time.

---

## [Author Response]

The following is the authors’ response to the current reviews.

**Reviewer #1 (Public review):**
Summary:Ferreiro et al. present a method to simulate protein sequence evolution under a birth-death model where sequence evolution is guided by structural constraints on protein stability. The authors then use this model to explore the predictability of sequence evolution in several viral proteins. In principle, this work is of great interest to molecular evolution and phylodynamics, which has struggled to couple non-neutral models of sequence evolution to phylodynamic models like birth-death processes. Unfortunately, though, the model shows little improvement over neutral models in predicting protein sequence evolution, although it can predict protein stability better than models assuming neutral evolution. It appears that more work is needed to determine exactly what aspects of protein sequence evolution are predictable under such non-neutral phylogenetic models.

We thank the reviewer for the positive comments about our work. We agree that further work is needed in the field of substitution models of molecular evolution to enable more accurate predictions of specific amino acid sequences in evolutionary processes.

Major concerns:(1) The authors have clarified the mapping between birth-death model parameters and fitness, but how fitness is modeled still appears somewhat problematic. The authors assume the death rate = 1 - birth rate. So a variant with a birth rate b = 1 would have a death rate d = 0 and so would be immortal and never die, which does not seem plausible. Also I'm not sure that this would "allow a constant global (birth-death) rate" as stated in line 172, as selection would still act to increase the population mean growth rate r = b - d. It seems more reasonable to assume that protein stability affects only either the birth or death rate and assume the other rate is constant, as in the Neher 2014 model.

The model proposed by Neher, et al. (2014), which incorporates a death rate (d) higher than 0 for any variant, was implemented and applied in the present method. In general, this model did not yield results different from those obtained using the model that assumes d = 1 – b, suggesting that this aspect may not be crucial for the study system. Next, the imposition of arbitrary death events based on an arbitrary death rate could be a point of concern. Regarding the original model, a variant with d = 0 can experience a decrease in fitness through the mutation process. In an evolutionary process, each variant is subject to mutation, and Markov models allow for the incorporation of mutations that decrease fitness (albeit with lower probability than beneficial ones, but they can still occur). All this information is included in the manuscript.

(2) It is difficult to evaluate the predictive performance of protein sequence evolution. This is in part due to the fact that performance is compared in terms of percent divergence, which is difficult to compare across viral proteins and datasets. Some protein sequences would be expected to diverge more because they are evolving over longer time scales, under higher substitution rates or under weaker purifying selection. It might therefore help to normalize the divergence between predicted and observed sequences by the expected or empirically observed amount of divergence seen over the timescale of prediction.

AU: The study protein datasets showed different levels of sequence divergence over their evolutionary times, as indicated for each dataset in the manuscript. For some metrics, we evaluated the accuracy (or error) of the predictions through direct comparisons between real and predicted protein variants using percentages to facilitate interpretation: 0% indicates a perfect prediction (no error), while 100% indicates a completely incorrect prediction (total error). Regarding normalization of these evaluations, we respectfully disagree with the suggestion because diverse factors can affect (not only the substitution rate, but also the sample size, structural features of the protein that may affect stability when accommodating different sequences, among others) and this complicates defining a consistent and meaningful normalization criterion. Given that the manuscript provides detailed information for each dataset, we believe that the presentation of the prediction accuracy through direct comparisons between real and predicted protein variants, expressed as percentages of similarity, is the clearest way.

(3) Predictability may also vary significantly across different sites in a protein. For example, mutations at many sites may have little impact on structural stability (in which case we would expect poor predictive performance) while even conservative changes at other sites may disrupt folding. I therefore feel that there remains much work to be done here in terms of figuring out where and when sequence evolution might be predictable under these types of models, and when sequence evolution might just be fundamentally unpredictable due to the high entropy of sequence space.

We agree with this reflection. Mutations can have different effects on folding stability, which are accounted for by the model presented in this study. However, accurately predicting the exact sequences of protein variants with similar stability remains difficult with current structurally constrained substitution models, and therefore, further work is needed in this regard. This aspect is indicated in the manuscript.

We want to thank the reviewer again for taking the time to revise our work and for the insightful and helpful comments.

**Reviewer #2 (Public review):**
In this study, the authors aim to forecast the evolution of viral proteins by simulating sequence changes under a constraint of folding stability. The central idea is that proteins must retain a certain level of structural stability (quantified by folding free energy, ΔG) to remain functional, and that this constraint can shape and restrict the space of viable evolutionary trajectories. The authors integrate a birth-death population model with a structurally constrained substitution (SCS) model and apply this simulation framework to several viral proteins from HIV-1, SARS-CoV-2, and influenza.The motivation to incorporate biophysical constraints into evolutionary models is scientifically sound, and the general approach aligns with a growing interest in bridging molecular evolution and structural biology. The authors focus on proteins where immune pressure is limited and stability is likely to be a dominant constraint, which is conceptually appropriate. The method generates sequence variants that preserve folding stability, suggesting that stability-based filtering may capture certain evolutionary patterns.

Correct. We thank the reviewer for the positive comments about our study.

However, the study does not substantiate its central claim of forecasting. The model does not predict future sequences with measurable accuracy, nor does it reproduce observed evolutionary paths. Validation is limited to endpoint comparisons in a few datasets. While KL divergence is used to compare amino acid distributions, this analysis is only applied to a single protein (HIV-1 MA), and there is no assessment of mutation-level predictive accuracy or quantification of how well simulated sequences recapitulate real evolutionary paths. No comparison is made to real intermediate variants available from extensive viral sequencing datasets which gather thousands of sequences with detailed collection date annotation (SARS-CoV-2, Influenza, RSV).

There are several points in this comment.

The presented method accurately predicts folding stability of forecasted variants, as shown through comparisons between real and predicted protein variants. However, as the reviewer correctly indicates, predicting the exact amino acid sequences remains challenging. This limitation is discussed in detail in the manuscript, where we also suggest that further improvements in substitution models of protein evolution are needed to better capture the evolutionary signatures of amino acid change at the sequence level, even between amino acids with similar physicochemical properties. Regarding the time points used for validation, the studied influenza NS1 dataset included two validation points. A key limitation in increasing the number of time points is the scarcity of datasets derived from monitoring protein evolution with sufficient molecular diversity between samples collected at consecutive time points (i.e., at least more than five polymorphic amino acid sites).

As described in the manuscript, calculating Kullback-Leibler (KL) divergence requires more than one sequence per studied time point. However, most datasets in the literature include only a single sequence per time point, typically a consensus sequence derived from bulk population sequencing. Generating multiple sequences per time point is experimentally more demanding, often requiring advanced methods such as single-virus sequencing or amplification of sublineages in viral subpopulations, as was done for the first dataset used in the study (Arenas, et al. 2016), which enabled the calculation of KL divergence. The extent to which the simulated sequences resemble real evolution is evaluated in the method validation. As noted, intermediate time point validation was performed using the influenza NS1 protein dataset. Although, as the reviewer indicates, thousands of viral sequences are available, these are usually consensus sequences from bulk sequencing. Indeed, many viral variants mainly differ through synonymous mutations, where the number of accumulated nonsynonymous mutations is small. For example, from the original Wuhan strain to the Omicron variant, the SARS-CoV-2 proteins Mpro and PLpro accumulated only 10 and 22 amino acid changes, respectively.

Analyzing intermediate variants of concern (i.e., Gamma or Delta) would reduce this number affecting statistics. In addition, many available viral sequences are not consecutive in evolutionary terms (one dataset does not represent the direct origin of another dataset at a subsequent time point), which further limits their applicability in this study. There is little data from monitored protein evolution with consecutive samples. The most suitable studies usually involve in vitro virus evolution, but the data from these studies often show low genetic variability between samples collected at different time points. Finally, it is important to note that the presented method can only be applied to proteins with known 3D structures, as it relies on selection based on folding stability. Non-structural proteins cannot be analyzed using this approach. Future work could incorporate additional selection constraints, which may improve the accuracy of predictions. These considerations and limitations are indicated in the manuscript.

The selection of proteins is narrow and the rationale for including or excluding specific proteins is not clearly justified.

The viral proteins included in the study were selected based on two main criteria, general interest and data availability. In particular, we included proteins from viruses that affect humans and for which data from monitored protein evolution, with sufficient molecular diversity between consecutive time points, is available. These aspects are indicated in the manuscript.

The analyzed datasets are also under-characterized: we are not given insight into how variable the sequences are or how surprising the simulated sequences might be relative to natural diversity. Furthermore, the use of consensus sequences to represent timepoints is problematic, particularly in the context of viral evolution, where divergent subclades often coexist - a consensus sequence may not accurately reflect the underlying population structure.

The manuscript indicates the sequence identity among protein datasets of different time points, along with other technical details. Next, the evaluation based on comparisons between simulated and real sequences reflects how surprising the simulated sequences might be relative to natural diversity, considering that the real dataset is representative. We believe that the diverse study real datasets are useful to evaluate the accuracy of the method in predicting different molecular patterns. Regarding the use of consensus sequences, we agree that they provide an approximation. However, as previously indicated, most of the available data from monitored protein evolution consist of consensus sequences obtained through bulk sequencing. Additionally, analyzing every individual viral sequence within a viral population, which is typically large, would be ideal but computationally intractable.

The fitness function used in the main simulations is based on absolute ΔG and rewards increased stability without testing whether real evolutionary trajectories tend to maintain, increase, or reduce folding stability over time for the particular systems (proteins) that are studied. While a variant of the model does attempt to center selection around empirical ΔG values, this more biologically plausible version is underutilized and not well validated.

The applied fitness function, based on absolute ΔG, is well stablished in the field (Sella and Hirsh 2005; Goldstein 2013). The present study independently predicts ΔG for the real and simulated protein variants at each sampling point. This ΔG prediction accounts not only for negative design, informed by empirical data, but also for positive design based on the study data (Arenas, et al. 2013; Minning, et al. 2013), thereby enabling the detection of variation in folding stability among protein variants. These aspects are indicated in the manuscript. Therefore, in our view, the study provides a proper comparison of real and predicted evolutionary trajectories in terms of folding stability.

Ultimately, the model constrains sequence evolution to stability-compatible trajectories but does not forecast which of these trajectories are likely to occur. It is better understood as a filter of biophysically plausible outcomes than as a predictive tool. The distinction between constraint-based plausibility and sequence-level forecasting should be made clearer. Despite these limitations, the work may be of interest to researchers developing simulation frameworks or exploring the role of protein stability in viral evolution, and it raises interesting questions about how biophysical constraints shape sequence space over time.

The presented method estimates the fitness of each protein variant, which can reflect the relative survival capacity of the variant. Therefore, despite the error due to evolutionary constraints not considered by the method, it indicates which variants are more likely to become fixed over time. In our view, the method does not merely filter plausible variants, rather, it generates predictions of variant survival through predicted fitness based on folding stability and simulations of protein evolution under structurally constrained substitution models integrated with birth-death population genetics approaches. The use of simulation-based approaches for prediction is well established in population genetics. For example, approaches such as approximate Bayesian computation (Beaumont, et al. 2002) rely on this strategy, and it has also been applied in other studies of forecasting evolution (e.g., Neher, et al. 2014). We believe that the distinction between forecasting folding stability and amino acid sequence is clearly shown in the manuscript, including the main text and the figures.

**Reviewer #2 (Recommendations for the authors):**
I thank the authors for addressing the question about template switching, their clarification was helpful. However, the core concerns I raised remain unresolved: the claim that the method is useful for forecasting is not substantiated. In order to support the paper's central claims or to prove its usefulness, several key improvements could be incorporated:(1) Systematic analysis of more proteins:The manuscript would be significantly strengthened by a systematic evaluation of model performance across a broader set of viral proteins, beyond the examples currently shown. Many human influenza and SARS-CoV-2 proteins have wellcharacterized structures or high-quality homology templates, making them suitable candidates. In the light of limited success of the method, presenting the model's behavior across a more comprehensive protein set, including those with varying structural constraints and immune pressures, would help assess generalizability and clarify the specific conditions under which the model is applicable.

Following a comment from the reviewer in a previous revision of the study, we included the analysis of an influenza NS1 protein dataset that contains two evaluation time points. Next, to validate the prediction method, it is necessary to have monitored protein sequences collected at least at two consecutive time points, with sufficient divergence between them to capture evolutionary signatures that allow for proper evaluation. Additionally, many data involve sequences that are not consecutive in evolutionary terms (one dataset is not a direct ancestor of another dataset existing at a posterior time point), which disallows their applicability in this study. Little data from monitored protein evolution with trustable consecutive (ancestor-descendant) samples exist. The most suitable studies often involve in vitro virus evolution, but they usually show low genetic variability between samples collected at different time points. Although thousands of sequences are available for some viruses, they are usually consensus sequences from bulk sequencing and often show a low number of nonsynonymous mutations at the study protein-coding gene between time points. For example, from the original Wuhan strain and the Omicron variant, the SARS-CoV-2 proteins Mpro and PLpro accumulated only 10 and 22 amino acid changes, respectively. Analyzing intermediate variants of concern (i.e., Gamma or Delta) would reduce this number affecting statistics. Thus, in practice, we found scarcity of data derived from monitoring protein evolution, with trustable ancestor and corresponding descendant data at consecutive time points and with sufficient molecular diversity between them (i.e., at least more than five polymorphic amino acid sites). In all, we believe that the diverse viral protein datasets used in the present study, along with the multiple analyzed datasets collected from monitored HIV-1 populations present in different patients, provide a representative application of the method, since notice that similar patterns were generally generated from the analysis of the different datasets.

(2) Present clear data statistics: For each analyzed dataset, the authors should provide basic information about the number of unique sequences, levels of variability, and evolutionary divergence between start and end sequences. This would contextualize the forecasting task and clarify whether the simulations are non-trivial. In particular, it should be shown that the consensus sequence is indeed representative of the viral population at a given time point. In viral evolution we frequently observe co-circulation of subclades and the consensus sequence is then not representative.

For each dataset analyzed, the manuscript provides the sequence identity between samples at the study time points (which also informs about sequence variability), sample sizes, representative protein structure, and other technical details. The study assumes that consensus sequences, typically generated by bulk sequencing, are representative of the viral population. Next, samples at different time points should involve ancestor-descendant relationships, which is a requirement and one of the limitations to find appropriate data for this study, as noted in our previous response.

(3) Explore other metrics for population level sequence comparison:In the light of possible existence of subclades, mentioned above, the currently used metrics for sequence comparison may underestimate performance of the simulations. It would be sufficient to see some overlap of simulated clades and the observed clades.

We found this to be a good idea. However, in practice, we believe that the criteria used to define subclades could introduce biases into the results. For some metrics, we evaluated the accuracy of the predictions through direct comparisons between all real and predicted protein variants, using percentages to facilitate interpretation. We believe that using subclades could potentially reduce the current prediction errors, but this would complicate the interpretation of the results, as they would be influenced by the subjective criteria used to define the subclades.

Currently, the manuscript presents a plausible filtering framework rather than a predictive model. Without these additional analyses, the main claims remain only partially supported.

Please see our reply to the comment of the reviewer just before the section titled “Recommendations for the authors”.

**Response to some rebuttal statements:**
(1) "Sequence comparisons based on the KL divergence require, at the studied time point, an observed distribution of amino acid frequencies among sites and an estimated distribution of amino acid frequencies among sites. In the study datasets, this is only the case for the HIV-1 MA dataset, which belongs to a previous study from one of us and collaborators where we obtained at least 20 independent sequences at each sampling point (Arenas, et al. 2016)"The available Influenza and SARS-CoV-2 data gathers isolates annotated with exact collection dates, providing reach datasets for such analysis.

The available influenza and SARS-CoV-2 sequences are typically derived from bulk sequencing and, therefore, they are consensus sequences. As a result, they cannot be used to calculate KL divergence. Additionally, many of the indicated sequences from databases are not demonstrated to be consecutive in evolutionary terms (one dataset is not a direct ancestor of another dataset existing at a posterior time point), which disallows their applicability in this study. The most suitable studies often involve in vitro virus evolution, but they usually show low genetic variability between samples collected at different time points.

(2) "Regarding extending the analysis to other time points (other variants of concern), we kindly disagree because Omicron is the variant of concern with the highest genetic distance to the Wuhan variant, and a high genetic distance is required to properly evaluate the prediction method."There have been many more variants of concern subsequent to Omicron which circulated in 2021.

A key aspect is the accumulation of diversity in the study proteins across different time points. The SARS-CoV-2 proteins Mpro and PLpro accumulated only 10 and 22 amino acid changes from the original Wuhan variant to Omicron, respectively.

Analyzing intermediate variants of concern (e.g., Gamma or Delta) or those closely related to Omicron would reduce the number of accumulated mutations even further.

We want to thank the reviewer again for taking the time to revise our work and for the insightful and helpful comments.

The following is the authors’ response to the original reviews.

**Reviewer #1 (Public review):**
Summary:Ferreiro et al. present a method to simulate protein sequence evolution under a birth-death model where sequence evolution is constrained by structural constraints on protein stability. The authors then use this model to explore the predictability of sequence evolution in several viral structural proteins. In principle, this work is of great interest to molecular evolution and phylodynamics, which have struggled to couple non-neutral models of sequence evolution to phylodynamic models like birth-death. Unfortunately, though, the model shows little improvement over neutral models in predicting protein evolution, and this ultimately appears to be due to fundamental conceptual problems with how fitness is modeled and linked to the phylodynamic birth-death model.

AU: We thank the reviewer for the positive comments about our work.

Regarding predictive power, the study showed a good accuracy in predicting the real folding stability of forecasted protein variants under a selection model, but not under a neutral model. Next, predicting the exact sequences was more challenging. In this revised version, where we added additional real data, we found that the accuracy of this prediction can vary among proteins (i.e., the SCS model was more accurate than the neutral model in predicting sequences of the influenza NS1 protein at different time points). Still, we consider that efforts are required in the field of substitution models of molecular evolution. For example, amino acids with similar physicochemical properties can result in predictions with appropriate folding stability while different specific sequence. The development of accurate substitution models of molecular evolution is an active area of research with ongoing progress, but further efforts are still needed. Next, forecasting the folding stability of future real proteins is fundamental for proper forecasting protein evolution, given the essential role of folding stability in protein function and its variety of applications. Regarding the conceptual concerns related to fitness modeling, we clarify them in detail in our responses to the specific comments below.

Major concerns:(1) Fitness model: All lineages have the same growth rate r = b-d because the authors assume b+d=1. But under a birth-death model, the growth r is equivalent to fitness, so this is essentially assuming all lineages have the same absolute fitness since increases in reproductive fitness (b) will simply trade off with decreases in survival (d). Thus, even if the SCS model constrains sequence evolution, the birthdeath model does not really allow for non-neutral evolution such that mutations can feed back and alter the structure of the phylogeny.

We thank the reviewer for this comment that aims to improve the realism of our model. In the model presented (but see later another model, derived from the proposal of the reviewer, that we have now implemented into the framework and applied it to the study data), the fitness predicted from a protein variant is used to obtain the corresponding birth rate of that variant. In this way, protein variants with high fitness have high birth rates leading to overall more birth events, while protein variants with low fitness have low birth rates resulting in overall more extinction events, which has biological meaning for the study system. The statement “All lineages have the same growth rate r = b-d” in our model is incorrect because, in our model, b and d can vary among lineages according to the fitness. For example, a lineage might have b=0.9, d=0.1, r=0.8, while another lineage could have b=0.6, d=0.4, r=0.2. Indeed, the statement “this is essentially assuming all lineages have the same absolute fitness” is incorrect. Clearly, assuming that all lineages have the same fitness would not make sense, in that situation the folding stability of the forecasted protein variants would be similar under any model, which is not the case as shown in the results. In our model, the fitness affects the reproductive success, where protein variants with a high fitness have higher birth rates leading to more birth events, while those with lower fitness have higher death rates leading to more extinction events. This parameterization is meaningful for protein evolution because the fitness of a protein variant can affect its survival (birth or extinction) without necessarily affecting its rate of evolution. While faster growth rate can sometimes be associated with higher fitness, a variant with high fitness does not necessarily accumulate substitutions at a faster rate. Regarding the phylogenetic structure, the model presented considers variable birth and death events across different lineages according to the fitness of the corresponding protein variants, and this affects the derived phylogeny (i.e., protein variants selected against can go extinct while others with high fitness can produce descendants). We are not sure about the meaning of the term “mutations can feed back” in the context of our system. Note that we use Markov models of evolution, which are well-stablished in the field (despite their limitations), and substitutions are fixed mutations, which still could be reverted later if selected by the substitution model (Yang 2006). Altogether, we find that the presented birth-death model is technically correct and appropriate for modeling our biological system. Its integration with structurally constrained substitution (SCS) models of protein evolution as Markov models follows general approaches of molecular evolution in population genetics (Yang 2006; Carvajal-Rodriguez 2010; Arenas 2012; Hoban, et al. 2012). We have now provided a more detailed description of the models in the manuscript.

Apart from these clarifications about the birth-death model used, we could understand the point of the reviewer and following the suggestion we have now incorporated an additional birth-death model that accounts for variable global birth-death rate among lineages. Specifically, we followed the model proposed by Neher et al (2014), where the death rate is considered as 1 and the birth rate is modeled as 1 + fitness. In this model, the global birth-death rate can vary among lineages. We implemented this model into the computer framework and applied it to the data used for the evaluation of the models. The results indicated that, in general, this model yields similar predictive accuracy compared to the previous birth-death model. Thus, accounting for variability in the global birth-death rate does not appear to play a major role in the studied systems of protein evolution. We have now presented this additional birth-death model and its results in the manuscript.

(2) Predictive performance: Similar performance in predicting amino acid frequencies is observed under both the SCS model and the neutral model. I suspect that this rather disappointing result owes to the fact that the absolute fitness of different viral variants could not actually change during the simulations (see comment #1).

As indicated in our previous answer, our study shows a good accuracy in predicting the real folding stability of forecasted protein variants under a selection model, but not under a neutral model. Next, predicting the exact sequences was more challenging, which was not surprising considering previous studies. In particular, inferring specific sequences is considerably challenging even for ancestral molecular reconstruction (Arenas, et al. 2017; Arenas and Bastolla 2020). Indeed, observed sequence diversity is much greater than observed structural diversity (Illergard, et al. 2009; Pascual-Garcia, et al. 2010), and substitutions between amino acids with similar physicochemical properties can yield modeled protein variants with more accurate folding stability, even when the exact amino acid sequences differ. As indicated, further work is demanded in the field of substitution models of molecular evolution. Next, in this revised version, where we included analyses of additional real datasets, we found that the accuracy of sequence prediction can vary among datasets. Notably, the analysis of an influenza NS1 protein dataset, with higher diversity than the other datasets studied, showed that the SCS model was more accurate than the neutral model in predicting sequences across different time points. Datasets with relatively high sequence diversity can contain more evolutionary information, which can improve prediction quality. In any case, as previously indicated, we believe that efforts are required in the field of substitution models of molecular evolution. Apart from that, forecasting the folding stability of future real proteins is an important advance in forecasting protein evolution, given the essential role of folding stability in protein function (Scheiblhofer, et al. 2017; Bloom and Neher 2023) and its variety of applications.

Next, also as indicated in our previous response, the birth-death model used in this study accounts for variation in fitness among lineages producing variable reproductive success. The additional birth-death model that we have now incorporated, which considers variation of the global birth-death rate among lineages, produced similar prediction accuracy, suggesting a limited role in protein evolution modeling. Molecular evolution parameters, particularly the substitution model, appear to be more critical in this regard. We have now included these aspects in the manuscript.

(3) Model assessment: It would be interesting to know how much the predictions were informed by the structurally constrained sequence evolution model versus the birth-death model. To explore this, the authors could consider three different models: (1) neutral, (2) SCS, and (3) SCS + BD. Simulations under the SCS model could be performed by simulating molecular evolution along just one hypothetical lineage. Seeing if the SCS + BD model improves over the SCS model alone would be another way of testing whether mutations could actually impact the evolutionary dynamics of lineages in the phylogeny.

In the present study, we compared the neutral model + birth-death (BD) with the SCS model + BD. Markov substitution models Q are applied upon an evolutionary time (i.e., branch length, t) and this allows to determine the probability of substitution events during that time period [P(t) = exp (Qt)]. This approach is traditionally used in phylogenetics to model the incorporation of substitution events over time. Therefore, to compare the neutral and SCS models in terms of evolutionary inference, an evolutionary time is required, in this case it is provided by the birth-death process. Thus, the cases (1) and (2) cannot be compared without an underlined evolutionary history. Next, comparisons in terms of likelihood, and other aspects, between models that ignore the protein structure and the implemented SCS models are already available in previous studies based on coalescent simulations or given phylogenetic trees (Arenas, et al. 2013; Arenas, et al. 2015). There, SCS models outperformed models that ignore evolutionary constraints from the protein structure, and those findings are consistent with the results obtained in the present study where we explored the application of these models to forecasting protein evolution. We would like to emphasize that forecasting the folding stability of future real proteins is a significant finding, folding stability is fundamental to protein function and has a variety of applications. We have now indicated these aspects in the manuscript.

(4) Background fitness effects: The model ignores background genetic variation in fitness. I think this is particularly important as the fitness effects of mutations in any one protein may be overshadowed by the fitness effects of mutations elsewhere in the genome. The model also ignores background changes in fitness due to the environment, but I acknowledge that might be beyond the scope of the current work.

AU: This comment made us realize that more information about the features of the implemented SCS models should be included in the manuscript. In particular, the implemented SCS models consider a negative design based on the observed residue contacts in nearly all proteins available in the Protein Data Bank (Arenas, et al. 2013; Arenas, et al. 2015). This data is distributed with the framework, and it can be updated to incorporate new structures (further details are provided in the distributed framework documentation and practical examples). Therefore, the prediction of folding stability is a combination of positive design (direct analysis of the target protein) and negative design (consideration of background proteins from a database to improve the predictions), thus incorporating background molecular diversity. We have now indicated this important aspect in the manuscript. Regarding the fitness caused by the environment, we agree with the reviewer. This is a challenge for any method aiming to forecast evolution, as future environmental shifts are inherently unpredictable and may affect the accuracy of the predictions. Although one might attempt to incorporate such effects into the model, doing so risks overparameterization, especially when the additional factors are uncertain or speculative. We have now mentioned this aspect in the manuscript.

(5) In contrast to the model explored here, recent work on multi-type birth-death processes has considered models where lineages have type-specific birth and/or death rates and therefore also type-specific growth rates and fitness (Stadler and Bonhoeffer, 2013; Kunhert et al., 2017; Barido-Sottani, 2023). Rasmussen & Stadler (eLife, 2019) even consider a multi-type birth-death model where the fitness effects of multiple mutations in a protein or viral genome collectively determine the overall fitness of a lineage. The key difference with this work presented here is that these models allow lineages to have different growth rates and fitness, so these models truly allow for non-neutral evolutionary dynamics. It would appear the authors might need to adopt a similar approach to successfully predict protein evolution.

We agree with the reviewer that robust birth-death models have been developed applying statistics and, in many cases, the primary aim of those studies is the development and refinement of the model itself. Regarding the study by Rasmussen and Stadler 2019, it incorporates an external evaluation of mutation events where the used fitness is specific for the proteins investigated in that study, which may pose challenges for users interested in analyzing other proteins. In contrast, our study takes a different approach. We implement a fitness function that can be predicted and evaluated for any type of structural protein (Goldstein 2013), making it broadly applicable. Actually, in this revised version we added the analysis of additional data of another protein (influenza NS1 protein) with predictions at different time points. In addition, we provide a freely available and well-documented computational framework to facilitate its use. The primary aim of our study is not the development of novel or complex birthdeath models. Rather, we aim to explore the integration of a standard birth-death model with SCS models for the purpose of predicting protein evolution. In the context of protein evolution, substitution models are a critical factor (Liberles, et al. 2012; Wilke 2012; Bordner and Mittelmann 2013; Echave, et al. 2016; Arenas, et al. 2017; Echave and Wilke 2017), and the presented combination with a birth-death model constitutes a first approximation upon which next studies can build to better understand this evolutionary system. We have now indicated these considerations in the manuscript.

**Reviewer #2 (Public review):**
Summary:In this study, "Forecasting protein evolution by integrating birth-death population models with structurally constrained substitution models", David Ferreiro and coauthors present a forward-in-time evolutionary simulation framework that integrates a birth-death population model with a fitness function based on protein folding stability. By incorporating structurally constrained substitution models and estimating fitness from ΔG values using homology-modeled structures, the authors aim to capture biophysically realistic evolutionary dynamics. The approach is implemented in a new version of their open-source software, ProteinEvolver2, and is applied to four viral proteins from HIV-1 and SARS-CoV-2.Overall, the study presents a compelling rationale for using folding stability as a constraint in evolutionary simulations and offers a novel framework and software to explore such dynamics. While the results are promising, particularly for predicting biophysical properties, the current analysis provides only partial evidence for true evolutionary forecasting, especially at the sequence level. The work offers a meaningful conceptual advance and a useful simulation tool, and sets the stage for more extensive validation in future studies.

We thank the reviewer for the positive comments on our study. Regarding the predictive power, the results showed good accuracy in predicting the folding stability of the forecasted protein variants. In this revised version, where we included analyses of additional real datasets, we found that the accuracy of sequence prediction can vary among datasets. Notably, the analysis of an influenza NS1 protein dataset, with higher diversity than the other datasets studied, showed that the SCS model was more accurate than the neutral model in predicting sequences across different time points. Datasets with relatively high sequence diversity can contain more evolutionary information, which can improve prediction quality. Still, we believe that further efforts are required in the field in improving the accuracy of substitution models of molecular evolution. Altogether, accurately forecasting the folding stability of future real proteins is fundamental for predicting their protein function and enabling a variety of applications. Also, we implemented the models into a freely available computer framework, with detailed documentation and a variety of practical examples.

Strengths:The results demonstrate that fitness constraints based on protein stability can prevent the emergence of unrealistic, destabilized variants - a limitation of traditional, neutral substitution models. In particular, the predicted folding stabilities of simulated protein variants closely match those observed in real variants, suggesting that the model captures relevant biophysical constraints.

We agree with the reviewer and appreciate the consideration that forecasting the folding stability of future real proteins is a relevant finding. For instance, folding stability is fundamental for protein function and affects several other molecular properties.

Weaknesses:The predictive scope of the method remains limited. While the model effectively preserves folding stability, its ability to forecast specific sequence content is not well supported.

Our study showed a good accuracy in predicting the real folding stability of forecasted protein variants under a selection model, but not under a neutral model. Predicting the exact sequences was more challenging, which was not surprising considering previous studies. In particular, inferring specific sequences is considerably challenging even for ancestral molecular reconstruction (Arenas, et al. 2017; Arenas and Bastolla 2020). Indeed, observed sequence diversity is much greater than observed structural diversity (Illergard, et al. 2009; Pascual-Garcia, et al. 2010), and substitutions between amino acids with similar physicochemical properties can yield modeled protein variants with more accurate folding stability, even when the exact amino acid sequences differ. As indicated, further work is demanded in the field of substitution models of molecular evolution. Next, in this revised version, where we included analyses of additional real datasets, we found that the accuracy of sequence prediction can vary among datasets. Notably, the analysis of an influenza NS1 protein dataset, with higher diversity than the other datasets studied, showed that the SCS model was more accurate than the neutral model in predicting sequences across different time points. Datasets with relatively high sequence diversity can contain more evolutionary information, which can improve prediction quality. In any case, as previously indicated, we believe that efforts are required in the field of substitution models of molecular evolution. Apart from that, forecasting the folding stability of future real proteins is an important advance in forecasting protein evolution, given the essential role of folding stability in protein function (Scheiblhofer, et al. 2017; Bloom and Neher 2023) and its variety of applications. We have now expanded these aspects in the manuscript.

Only one dataset (HIV-1 MA) is evaluated for sequence-level divergence using KL divergence; this analysis is absent for the other proteins. The authors use a consensus Omicron sequence as a representative endpoint for SARS-CoV-2, which overlooks the rich longitudinal sequence data available from GISAID. The use of just one consensus from a single time point is not fully justified, given the extensive temporal and geographical sampling available. Extending the analysis to include multiple timepoints, particularly for SARS-CoV-2, would strengthen the predictive claims. Similarly, applying the model to other well-sampled viral proteins, such as those from influenza or RSV, would broaden its relevance and test its generalizability.

The evaluation of forecasting evolution using real datasets is complex due to several conceptual and practical aspects. In contrast to traditional phylogenetic reconstruction of past evolutionary events and ancestral sequences, forecasting evolution often begins with a variant that is evolved forward in time and requires a rough fitness landscape to select among possible future variants (Lässig, et al. 2017). Another concern for validating the method is the need to know the initial variant that gives rise to the corresponding future (forecasted) variants, and it is not always known. Thus, we investigated systems where the initial variant, or a close approximation, is known, such as scenarios of in vitro monitored evolution. In the case of SARS-CoV-2, the Wuhan variant is commonly used as the starting variant of the pandemic. Next, since forecasting evolution is highly dependent on the used model of evolution, unexpected external factors can be dramatic for the predictions. For this reason, systems with minimal external influences provide a more controlled context for evaluating forecasting evolution. For instance, scenarios of in vitro monitored virus evolution avoid some external factors such as host immune responses. Another important aspect is the availability of data at two (i.e., present and future) or more time points along the evolutionary trajectory, with sufficient genetic diversity between them to identify clear evolutionary signatures. Additionally, using consensus sequences can help mitigate effects from unfixed mutations, which should not be modeled by a substitution model of evolution. Altogether, not all datasets are appropriate to properly evaluate or apply forecasting evolution. These aspects are indicated in the manuscript. Sequence comparisons based on the KL divergence require, at the studied time point, an observed distribution of amino acid frequencies among sites and an estimated distribution of amino acid frequencies among sites. In the study datasets, this is only the case for the HIV-1 MA dataset, which belongs to a previous study from one of us and collaborators where we obtained at least 20 independent sequences at each sampling point (Arenas, et al. 2016). This aspect is now more clearly indicated in the manuscript. Regarding the Omicron datasets, we used 384 curated sequences of the Omicron variant of concern to construct the study data and we believe that it is a representative sample. The sequence used for the initial time point was the Wuhan variant (Wu, et al. 2020), which is commonly assumed to be the origin of the pandemic in SARS-CoV-2 studies. As previously indicated, the use of consensus sequences is convenient to avoid variants with unfixed mutations. Regarding extending the analysis to other time points (other variants of concern), we kindly disagree because Omicron is the variant of concern with the highest genetic distance to the Wuhan variant, and a high genetic distance is required to properly evaluate the prediction method. Actually, we noted that earlier variants of concern show a small number of fixed mutations in the study proteins, despite the availability of large numbers of sequences in databases such as GISAID. Additionally, we investigated the evolutionary trajectories of HIV-1 protease (PR) in 12 intra-host viral populations with predictions for up to four different time points. Apart from those aspects, following the proposal of the reviewer, we have now incorporated the analysis of an additional dataset of influenza NS1 protein (Bao, et al. 2008), with predictions for two different time points, to further assess the generalizability of the method. We have now included details of this influenza NS1 protein dataset and the predictions derived from it in the manuscript.

It would also be informative to include a retrospective analysis of the evolution of protein stability along known historical trajectories. This would allow the authors to assess whether folding stability is indeed preserved in real-world evolution, as assumed in their model.

Our present study does not aim to investigate the evolution of the folding stability over time, although it provides this information indirectly at the studied time points. Instead, the present study shows that the folding stability of the forecasted protein variants is similar to the folding stability of the corresponding real protein variants for diverse viral proteins, which provides an important evaluation of the prediction method. Next, the folding stability can indeed vary over time in both real and modeled evolutionary scenarios, and our present study is not in conflict with this. In that regard, which is not the aim of our present study, some previous phylogenetic-based studies have reported temporal fluctuations in folding stability for diverse protein data (Arenas, et al. 2017; Olabode, et al. 2017; Arenas and Bastolla 2020; Ferreiro, et al. 2022).

Finally, a discussion on the impact of structural templates - and whether the fixed template remains valid across divergent sequences - would be valuable. Addressing the possibility of structural remodeling or template switching during evolution would improve confidence in the model's applicability to more divergent evolutionary scenarios.

This is an important point. For the datasets that required homology modeling (in several cases it was not necessary because the sequence was present in a protein structure of the PDB), the structural templates were selected using SWISS-MODEL, and we applied the best-fitting template. We have now included in a supplementary table details about the fitting of the structural templates. Indeed, our proposal assumes that the protein structure is maintained over the studied evolutionary time, which can be generally reasonable for short timescales where the structure is conserved (Illergard, et al. 2009; Pascual-Garcia, et al. 2010). Over longer evolutionary timescales, structural changes may occur and, in such cases, modeling the evolution of the protein structure would be necessary. To our knowledge, modeling the evolution of the protein structure remains a challenging task that requires substantial methodological developments. Recent advances in artificial intelligence, particularly in protein structure prediction from sequence, may offer promising tools for addressing this challenge. However, we believe that evaluating such approaches in the context of structural evolution would be difficult, especially given the limited availability of real data with known evolutionary trajectories involving structural change. In any case, this is probably an important direction for future research. We have now included this discussion in the manuscript.

**Reviewer #1 (Recommendations for the authors):**
(1) Abstract: "expectedly, the errors grew up in the prediction of the corresponding sequences" <- Not entirely clear what is meant by "errors grew up" or what the errors grew with.

This sentence refers to the accuracy of sequence prediction in comparison to that of folding stability prediction. We have now clarified this aspect in the manuscript.

(2) Lines 162-165: "Alternatively, if the fitness is determined based on the similarity in folding stability between the modeled variant and a real variant, the birth rate is assumed to be 1 minus the root mean square deviation (RMSD) in folding stability." <- What is the biological motivation for using the RMSD? It seems like a more stable variant would always have higher fitness, at least according to Equation 1.

RMSD is commonly used in molecular biology to compare proteins in terms of structural distance, folding stability, kinetics, and other properties. It offers advantages such as minimizing the influence of small deviations while amplifying larger differences, thereby enhancing the detection of remarkable molecular changes. Additionally, RMSD would facilitate the incorporation of other biophysical parameters, such as structural divergences from a wild-type variant or entropy, which could be informative for fitness in future versions of the method. We have now included this consideration in the manuscript.

(3) Lines 165-166: "In both cases, the death rate (d) is considered as 1-b to allow a constant global (birth-death) rate" <- This would give a constant R = b / (1-b) over the entire phylogenetic tree. For applications to pathogens like viruses with epidemic dynamics, this is extremely implausible. Is there any need to make such a restrictive assumption?

Regarding technical considerations of the model, we refer to our answer to the first public review comment. Next, a constant global rate of evolution was observed in numerous genes and proteins of diverse organisms, including viruses (Gojobori, et al.1990; Leitner and Albert 1999; Shankarappa, et al. 1999; Liu, et al. 2004; Lu, et al. 2018; Zhou, et al. 2019). However, following the comment of the reviewer, and as we indicated in our answer to the first public review comment, we have now implemented and evaluated an additional birth-death model that allows for variation in the global birth-death rate among lineages. We have implemented this additional model in the framework and described it along with its results in the manuscript.

(4) Lines 187-188: "As a consequence, since b+d=1 at each node, tn is consistent across all nodes, according to (Harmon, 2019)." <- This would also imply that all lineages have a growth rate r = b - d, which under a birth-death model is equivalent to saying all lineages have the same fitness!

We clarified this aspect in our answer to the first public review comment. In particular, in the model presented, protein variants with higher fitness have higher birth rates, leading to more birth events, while protein variants with lower fitness have lower birth rates leading to more extinction events, which presents biological meaning for the study system. In our model b and d can vary among lineages according to the corresponding fitness (i.e., a lineage may have b=0.9, d=0.1, r=0.8; while another one may have b=0.6, d=0.4, r=0.2). Since the reproductive success varies among lineages in our model, the statement “this is essentially assuming all lineages have the same absolute fitness” is incorrect, although it could be interpreted like that in certain models. Fitness affects reproductive success, but fitness and growth rate of evolution are different biological processes (despite a faster growth rate can sometimes be associated with higher fitness, a variant with a high fitness not necessarily has to accumulate substitutions at a higher rate). An example in molecular adaptation studies is the traditional nonsynonymous to synonymous substitution rates ratio (dN/dS), where dN/dS (that informs about selection derived from fitness) can be constant at different rates of evolution (dN and dS). In any case, we thank the reviewer for raising this point, which led us to incorporate an additional birth-death model and inspired some ideas. Thus, following the comment of the reviewer and as indicated in the answer to the first public review comment, we have now implemented and evaluated an additional birthdeath model that allows for variation in the global birth-death rate among lineages. The results indicated that this model yields similar predictive accuracy compared to the previous birth-death model. We have now included these aspects, along with the results from the additional model, in the manuscript.

(5) Line 321-322: "For the case of neutral evolution, all protein variants equally fit and are allowed, leading to only birth events," <- Why would there only be birth events? Lineages can die regardless of their fitness.

AU: In the neutral evolution model, all protein variants have the same fitness, resulting in a flat fitness landscape. Since variants are observed, we allowed birth events. However, it assumed the absence of death events as no information independent of fitness is available to support their inclusion and quantification, thereby avoiding the imposition of arbitrary death events based on an arbitrary death rate. We have now provided a justification of this assumption in the manuscript.

**Reviewer #2 (Recommendations for the authors):**
(1) Clarify the purpose of the alternative fitness mode ("ΔG similarity to a target variant"):The manuscript briefly introduces an alternative fitness function based on the similarity of a simulated protein's folding stability to that of a real protein variant, but does not provide a clear motivation, usage scenario, or results derived from it.

The presented model provides two approaches for deriving fitness from predicted folding stability. The simpler approach assumes that a more stable protein variant has higher fitness than a less stable one. The alternative approach assigns high fitness to protein variants whose stability closely matches observed stability, acknowledging that the real observed stability is derived from the real selection process, and this approach considers negative design by contrasting the prediction with real information. For the analyses of real data in this study, we used the second approach, guided by these considerations. We have now clarified this aspect in the manuscript.

(2) Report structural template quality and modeling confidence:Since folding stability (ΔG) estimates rely on structural models derived from homology templates, the accuracy of these predictions will be sensitive to the choice and quality of the template structure. I recommend that the authors report, for each protein modeled, the template's sequence identity, coverage, and modeling quality scores. This will help readers assess the confidence in the ΔG estimates and interpret how template quality might impact simulation outcomes.

We agree with the reviewer and we have now included additional information in a supplementary table regarding sequence identity, modeling quality and coverage of the structural templates for the proteins that required homology modeling. The selection of templates was performed using the well-established framework SWISS-MODEL and the best-fitting template was chosen. Next, a large number of protein structures are available in the PDB for the study proteins, which favors the accuracy of the homology modeling. For some datasets, homology modeling was not required, as the modeled sequence was already present in an available protein structure. We have now included this information in the manuscript and in a supplementary table.

(3) Clarify whether structural remodeling occurs during simulation:It appears that folding stability (ΔG) for all simulated protein variants is computed by mapping them onto a single initial homology model, without remodeling the structure as sequences evolve. If correct, this should be clearly stated, as it assumes that the structural fold remains valid across all simulated variants. A discussion on the potential impact of structural drift would be welcome.

We agree with the reviewer. As indicated in our answer to a previous comment, our method assumes that the protein structure is maintained over the studied evolutionary time, which is generally acceptable for short timescales where the structure is conserved (Illergard, et al. 2009; Pascual-Garcia, et al. 2010). At longer timescales the protein structure could change, requiring the modeling of structural evolution over the evolutionary time. To our knowledge, modeling the evolution of the protein structure remains a challenging task that requires substantial methodological developments. Recent advances in artificial intelligence, particularly in protein structure prediction from sequence, can be promising tools for addressing this challenge. However, we believe that evaluating such approaches in the context of structural evolution would be difficult, especially given the limited availability of real datasets with known evolutionary trajectories involving structural change. In any case, this is probably an important direction for future research. We have now included this discussion in the manuscript.